# The components of an electrical synapse as revealed by expansion microscopy of a single synaptic contact

Sandra P Cárdenas-García, Sundas Ijaz, Alberto E Pereda*

Dominick P. Purpura Department of Neuroscience, Albert Einstein College of Medicine, Bronx, United States

**Abstract** Most nervous systems combine both transmitter-mediated and direct cell–cell communication, known as 'chemical' and 'electrical' synapses, respectively. Chemical synapses can be identified by their multiple structural components. Electrical synapses are, on the other hand, generally defined by the presence of a 'gap junction' (a cluster of intercellular channels) between two neuronal processes. However, while gap junctions provide the communicating mechanism, it is unknown whether electrical transmission requires the contribution of additional cellular structures. We investigated this question at identifiable single synaptic contacts on the zebrafish Mauthner cells, at which gap junctions coexist with specializations for neurotransmitter release and where the contact unequivocally defines the anatomical limits of a synapse. Expansion microscopy of these single contacts revealed a detailed map of the incidence and spatial distribution of proteins pertaining to various synaptic structures. Multiple gap junctions of variable size were identified by the presence of their molecular components. Remarkably, most of the synaptic contact's surface was occupied by interleaving gap junctions and components of adherens junctions, suggesting a close functional association between these two structures. In contrast, glutamate receptors were confined to small peripheral portions of the contact, indicating that most of the synaptic area functions as an electrical synapse. Thus, our results revealed the overarching organization of an electrical synapse that operates with not one, but multiple gap junctions, in close association with structural and signaling molecules known to be components of adherens junctions. The relationship between these intercellular structures will aid in establishing the boundaries of electrical synapses found throughout animal connectomes and provide insight into the structural organization and functional diversity of electrical synapses.

*For correspondence:
alberto.pereda@einsteinmed.
edu

**Competing interest:** The authors declare that no competing interests exist.

## Editor's evaluation

This manuscript provides fundamental insights into how components of an electrical synapse are arranged at identified gap junctions using expansion microscopy. They provide convincing evidence for how these molecular components are placed within the junction. Such analysis is important for our understanding of synaptic organization and function.

## Introduction

Synapses are specialized cell–cell contacts where two neurons can share relevant functional information. The exchange of information can occur directly through cell–cell channels, known as 'electrical synapses', or indirectly, via the release of a chemical messenger, known as 'chemical synapses' (*Pereda, 2014*). While the molecular complexity of chemical synapses with structurally distinct pre- and postsynaptic components has long been recognized (*Wichmann and Kuner, 2022*; *Wilhelm*

**eLife digest** Neurons communicate with each other through specialized structures known as synapses. At chemical synapses, the cells do not physically interact as they rely instead on molecules called neurotransmitters to pass along signals. At electrical synapses, however, neurons are directly connected via gap junctions, which are clusters of intercellular channels that allow ions and other small compounds to move from one cell to another.

Both electrical and chemical synapses play critical roles in neural circuits, and both exhibit some amount of plasticity – they weaken or strengthen depending on how often they are used, an important feature for the brain to adapt to the needs of the environment. Yet the structure and molecular organization of electrical synapses have remained poorly understood compared to their chemical counterparts.

In response, Cárdenas-García, Ijaz and Pereda took advantage of a new approach known as expansion microscopy to examine the electrical synapse that connects neurons bringing sound information to a pair of unusually large neurons in the brain of most bony fish. With this method, a biological sample is prepared in such a way that its size increases, but the relative position of its components is preserved. This allows scientists to better observe structures that would otherwise be too difficult to capture using traditional microscopy techniques.

Experiments in larval zebrafish revealed that contrary to previous assumptions, the electrical synapse was formed of not one but multiple gap junctions of various sizes closely associated with a range of structural and signaling molecules typically found in adherens junctions (a type of structure that physically links cells together). The team suggests that these molecular actors could work to ensure that the multiple gap junctions act in concert at the synapse. Overall, these findings offer a new perspective on how electrical synapses are organized and regulated, which refines our understanding of how the nervous system functions both in health and in disease.

---

*et al., 2014*), less is known regarding the molecular and structural complexity of electrical synapses. Electrical synapses are a modality of neuronal communication mediated by structures known as 'gap junctions' (GJs) (*Goodenough and Paul, 2009*). These structures contain intercellular channels formed by the apposition of two hemichannels, each provided by one of the connected cells, and which cluster together into GJ 'plaques' (*Goodenough and Paul, 2009*). Hemichannels are formed by proteins called 'connexins' (*Goodenough and Paul, 2009*; *Söhl and Willecke, 2004*) in vertebrates and 'innexins' (*Phelan et al., 1998*; *Phelan, 2005*) in invertebrates that, though unrelated in sequence, share a similar membrane topology that allows them to assemble into intercellular channels. While GJs are ubiquitous and present in virtually every tissue of the organism providing metabolic coupling (*Goodenough and Paul, 2009*), they additionally serve as a pathway of low resistance for the spread of electrical currents between neurons (and cells of the heart), the main form of signaling in the brain, which is fast enough to operate within the time frame required for decision-making by neural circuits (*Bennett, 1997*; *Alcamí and Pereda, 2019*). Electrical synapses are generally perceived as structurally simpler than chemical synapses and exclusively involving the function of intercellular channels. However, recent data indicates that the function of these channels is under the control of their supporting molecular scaffold (*Flores et al., 2008*; *Lasseigne et al., 2021*; *Martin et al., 2020*; *Martin et al., 2023*), suggesting that neuronal GJs are complex molecular structures whose function requires the contribution of multiple molecular components (*Miller and Pereda, 2017*). Such molecular complexity is likely to underlie plastic changes in the strength of electrical synapses (*Pereda et al., 2013*; *O'Brien and Bloomfield, 2018*), which are capable of dynamically reconfiguring neural circuits (*O'Brien and Bloomfield, 2018*; *Bloomfield and Völgyi, 2009*).

Thus far, investigations of the properties of vertebrate electrical transmission have solely considered the functional properties of the channel-forming proteins, the connexins, the molecules that regulate them, and their interactions at the GJ plaque. However, could a single neuronal GJ per se be considered an electrical synapse? Alternatively, does electrical transmission rely on additional structural components? The identification of the structural components of a chemical synapse is facilitated by the presynaptic bouton, which anatomically defines its limit (*Sotelo, 2020*). In contrast, neuronal GJs are typically found connecting cell somata or other neuronal processes (*Alcamí and Pereda,*

*2019*; *Sotelo, 2020*; *Sotelo and Llinás, 1972*), such as dendrites and axons, making it more difficult to define the exact anatomical boundaries that constitute an electrical synapse. Neuronal GJs can also occur at synaptic boutons, generally coexisting with specializations for chemical transmission (*Sotelo, 2020*; *Martin and Pilar, 1963*). This is the case for auditory afferents terminating as single 'large myelinated club endings' (*Bartelmez, 1915*; *Bartelmez and Hoerr, 1933*) or 'club endings' (CEs) on the lateral dendrite of the teleost Mauthner cells (a pair of large reticulospinal neurons involved in tail-flip escape responses in fish) (*Pereda and Faber, 2011*; *Faber and Pereda, 2011*), each containing GJs and specializations for chemical transmission (*Robertson et al., 1963*). Because of their experimental accessibly and functional properties, these terminals are considered a valuable model to study vertebrate electrical transmission as they more easily allow for the correlation between structure and function of synaptic features (*Pereda et al., 2013*; *Robertson et al., 1963*; *Furshpan, 1964*; *Flores et al., 2012*). Since the bouton marks the anatomical limits of a synapse, these contacts offer the opportunity to examine the anatomical structures that together make an electrical synapse.

Here, we used expansion microscopy (*Tillberg et al., 2016*) to expose the presence and spatial arrangement of synaptic components in CEs of larval zebrafish. CEs from larval zebrafish share comparable morphological and functional properties with those of adult fish (*Lasseigne et al., 2021*; *Yao et al., 2014*; *Miller et al., 2017*; *Echeverry et al., 2022*), and due to their genetic accessibility, allow for the opportunity to investigate the functional link between the structures enabling electrical transmission and its regulation. Expansion revealed the presence of multiple well-defined puncta distributed throughout the contact area, which, consistent with the notion that they represent GJ plaques, each exhibited labeling for the fish homologs of the widespread mammalian connexin 36 (Cx36), Cx35, and Cx34, as well as for the GJ scaffolding protein zonula occludens (ZO1) (*Flores et al., 2008*; *Lasseigne et al., 2021*; *Miller et al., 2017*; *Rash et al., 2013*). Strikingly, expansion following staining with N-cadherin and ß-catenin antibodies, protein components of 'adherens junctions' (AJs), showed that these proteins are also distributed all throughout the contact area, but in a fashion that is mutually exclusive with connexin. This suggests that the subsynaptic topography of electrical synapses is carefully and intimately coordinated. Finally, double labeling with Cx and glutamate receptor antibodies showed that, while GJs are distributed throughout the entire contact area, a much smaller number of glutamatergic sites are restricted to the periphery occupying a small fraction (~19%) of the contact's surface. Thus, our data suggest that synaptic communication at electrical synapses results from not one but the coordinated action of multiple GJs of variable size, which may require the functional contribution of additional structures, such as AJs.

## Results

A group of auditory afferents, each of which terminate as a single synaptic contact, known as a club ending (CE), on the distal portion of the lateral dendrite of the Mauthner (M-) cell (*Bartelmez, 1915*; *Bartelmez and Hoerr, 1933*; *Figure 1A*). Because of their unusual large size and experimental accessibility, CEs represent a valued model for the correlation of synaptic structure and function. Ultrastructural analysis of CEs in adult goldfish (*Robertson et al., 1963*; *Tuttle et al., 1986*; *Kohno and Noguchi, 1986*) and larval zebrafish (*Yao et al., 2014*) revealed the presence of GJs coexisting with specializations for neurotransmitter release. Consistent with these synaptic specializations, stimulation of CEs evokes a synaptic response that combines electrical and chemical transmission (*Furshpan, 1964*; *Yao et al., 2014*; *Echeverry et al., 2022*; *Lin and Faber, 1988*). A wealth of evidence indicates that GJs at these terminals consist of heterotypic intercellular channels created by the apposition of a presynaptic hemichannel formed by connexin Cx35.5 and a postsynaptic hemichannel formed by Cx34.1, two of the Cx35 (Cx35.1 and Cx35.5) and Cx34 (Cx34.1 and Cx34.7) orthologs (*Miller et al., 2017*; *Rash et al., 2013*). These junctions also contain the scaffolding protein ZO1 (*Flores et al., 2008*; *Lasseigne et al., 2021*), which regulates channel function. The synaptic contact areas of CEs (*Figure 1A*) can be visualized and unambiguously identified by immunolabeling for these GJ proteins, which are revealed as large fluorescent oval areas at the distal portion of the lateral dendrite of the M-cell (*Figure 1B–D*).

Immunolabeling is commonly used to define the biochemical composition of a synapse by allowing detection of the presence of specific proteins. We combined this approach with a protein-retention expansion microscopy protocol (proExM; *Tillberg et al., 2016*) to explore not only the presence but the relative distribution of the structures formed by various synaptic proteins throughout the contact

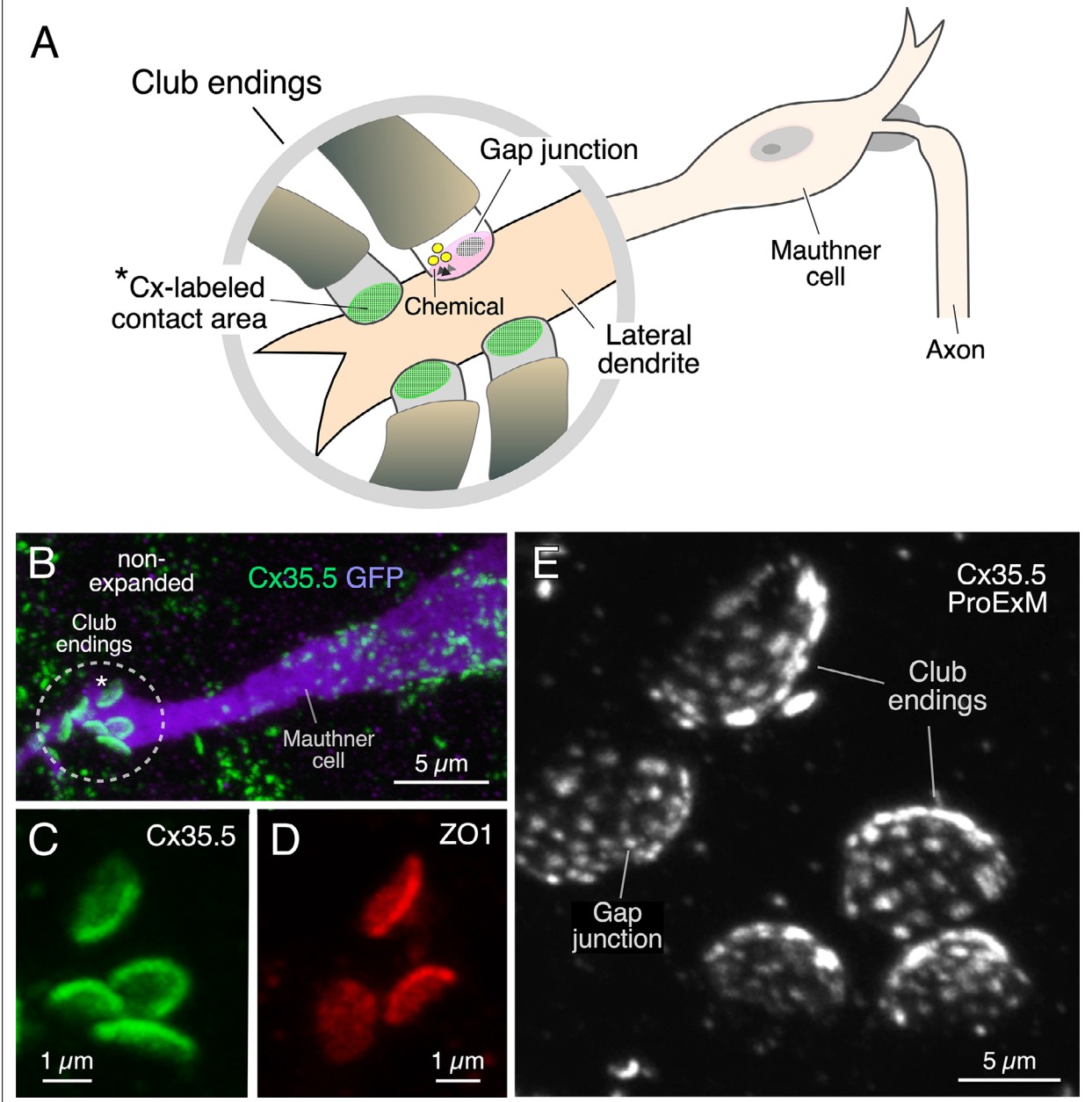

**Figure 1.** Expansion microscopy of club ending (CE) contact areas in larval zebrafish. (**A**) The cartoon illustrates the auditory afferents that terminate as single CEs, each containing gap junctions (GJs, green) and specializations for chemical transmission (Chemical), on the distal portion of the lateral dendrite of the Mauthner (M-) cell. Synaptic contact areas labeled with connexin antibody (see **B** and **C**) are represented in green. (**B**) Confocal image with anti-GFP (purple) and anti-Cx35/36 (green), which labels both Cx35.5 and Cx35.1, showing a long stretch of the lateral dendrite of the M-cell (projection of 34 confocal z-sections at 0.39 µm z-step size), revealing the contact areas (indicated by the asterisk here and the cartoon of **A**) of several CEs. (**C, D**) Contact areas of individual CEs labeled with anti-Cx35/36 (**C**, green; projection of 12 sections at 0.39 µm z-step size) and anti-ZO1 (**D**, red; projection of 4 sections at 0.39 µm z-step size). (**E**) Protein-retention expansion microscopy (ProExM) with anti-Cx35.5 increases the size of CE synaptic contact areas, enabling the visualization of intrasynaptic components (projection of 19 sections at 0.88 µm z-step size). The scale bars represent actual dimensions and, therefore, ProExM images were not adjusted for expansion factor.

The online version of this article includes the following video and figure supplement(s) for figure 1:

**Figure supplement 1.** Short–long-diameter ratio probability plots.

**Figure 1—video 1.** Expansion microscopy with anti-Cx35.5.

https://elifesciences.org/articles/91931/figures#fig1video1

areas of CEs in 5 days post fertilization (dpf) zebrafish. Tissue expansion with anti-Cx35.5 revealed the presence of multiple Cx35.5-positive puncta at these contacts forming a concave oval area (*Figure 1E*, *Figure 1—video 1*), reminiscent of that observed at goldfish CEs, known to correspond to GJ plaques (*Flores et al., 2008*; *Pereda et al., 2003*). The main diameter of the oval synaptic contact area defined by labeling of Cx35.5, either using anti-Cx35/36 (which recognizes both Cx35.5 and Cx35.1) or a specific anti-Cx35.5, or of ZO1 with anti-ZO1, increased about fourfold (3.9× expansion factor), from 2.11 ± 0.037 µm in non-expansion experiments (n = 38 CEs from 26 fish) to 8.25 ± 0.225 µm in expansion experiments (n = 40 CEs from 13 fish) (mean ± SEM). Moreover, expansion led to an about 13-fold increase in the area of the contact (13.4×), from 2.63 ± 0.055 µm$^2$ (n = 38 CEs from 26 fish) to 35.13 ± 1.266 µm$^2$ (n = 40 CEs from 13 fish). A concern when using proExM is whether this procedure leads to the distortion of normal anatomical features. Therefore, to determine the degree of isotropy of the expansion, we measured the ratio between the short and long diameters of the CE oval areas (S/L ratio) in non- and post-expanded samples. The short and long diameters in non-expanded tissue averaged 1.62 ± 0.026 µm and 2.11 ± 0.037 µm (n = 38 CEs from 26 fish), respectively, and 5.16 ± 0.142 µm and 8.25 ± 0.225 µm (n = 40 CEs from 13 fish), respectively, in expanded terminals. Analysis showed that the average S/L ratio decreased in the expanded tissue, with the decrease reaching statistical significance (non-expanded, 0.77 ± 0.018 [n = 38 CEs from 26 fish]; expanded, 0.64 ± 0.019 [n = 40 CEs from 13 fish]; p<0.01). However, the decrease in the average S/L ratio was less than 18%. Moreover, the distribution of S/L ratios from expanded tissue was nearly identical to the distribution from non-expanded tissue, and both were well described by normal distributions (*Figure 1—figure supplement 1*). This result indicates that the expansion procedure had no selective effects on CEs, but rather affected all uniformly. The difference between expanded and non-expanded S/L ratios might also result from an underestimation of the small diameter due to a minor tilting along the long diameter in 'en face' views of expanded CEs, which are more difficult to obtain because of their larger size. (In contrast to adult animals, 'en face' views are harder to obtain in larval zebrafish because the diameter of the CE contact in larvae is comparable to the diameter of the lateral dendrite of the M-cell [*Yao et al., 2014*], which, together with the presence of a smaller number of these afferents in larvae, makes 'en face' views less likely to be found and measured.) Together, with our synaptic alignment findings (see below), these results show that the expansion procedure had minimal effects on synaptic structure. Thus, expansion of these single synapses resulted in a more than tenfold increase of the synaptic contact area, allowing for a more detailed visualization of the relative distribution of its synaptic components.

CEs combine electrical with chemical transmission. To investigate the area occupied by each form of transmission, we labeled for Cx35.5 as a marker of GJs and glutamate receptor 2 (GluR2) as a marker of glutamatergic transmission (*Figure 2A–C*). As expected, colocalization analysis (Manders' coefficient analysis) revealed that labeling for Cx35.5 and GluR2 was mutually exclusive (*Figure 2D*). Yet, while Cx35.5 labeling covered the majority of the area, labeling for GluR2 was substantially lower and limited to the contact's peripheral margin (*Figure 2A–C*). To quantify this differential distribution, we defined two regions of interest (ROIs) at 'en face' views of the CE contact: a central, oval ROI representing ¾ of the area ('center') and an annular ROI representing the peripheral, remaining ¼ of the area ('periphery'). Estimates of the relative intensity of Cx35.5 vs. GluR2 in each ROI showed that, while GluR2 is constrained to the periphery, Cx35.5 is homogeneously distributed through the contact area (*Figure 2E and F*; see figure legend and 'Materials and methods' section for statistical analysis). This distribution is consistent with previous EM reconstruction of CEs in adult goldfish (*Tuttle et al., 1986*), at which specializations for transmitter release were found to be restricted to the periphery of the contact. Thus, while GJs occupy most of the surface, chemical transmission occupies a smaller and peripheral portion, suggesting that most of the contact operates as an electrical synapse.

Rather than diffusely distributed, labeling for Cx35.5 in expansion samples was characterized by well-defined puncta distributed throughout the contact area, suggesting that they each might represent an individual GJ plaque. Consistent with this interpretation, double labeling for Cx35.5 and Cx34.1 (*Figure 3A and C*) showed a high degree of colocalization at CE contact areas (*Figure 3E*). A high degree of colocalization (*Figure 3B and D*, *Figure 3—video 1*) was also found between labeling for Cx35.5 and ZO1 (*Figure 3F*; see figure legend for statistical analysis). Moreover, labeling for Cx35.5 and Cx34.1 colocalized at single puncta, as indicated by line scan of individual puncta. Given the characteristic concavity of the CE contact area, the relationship between labeled pre- and postsynaptic GJ

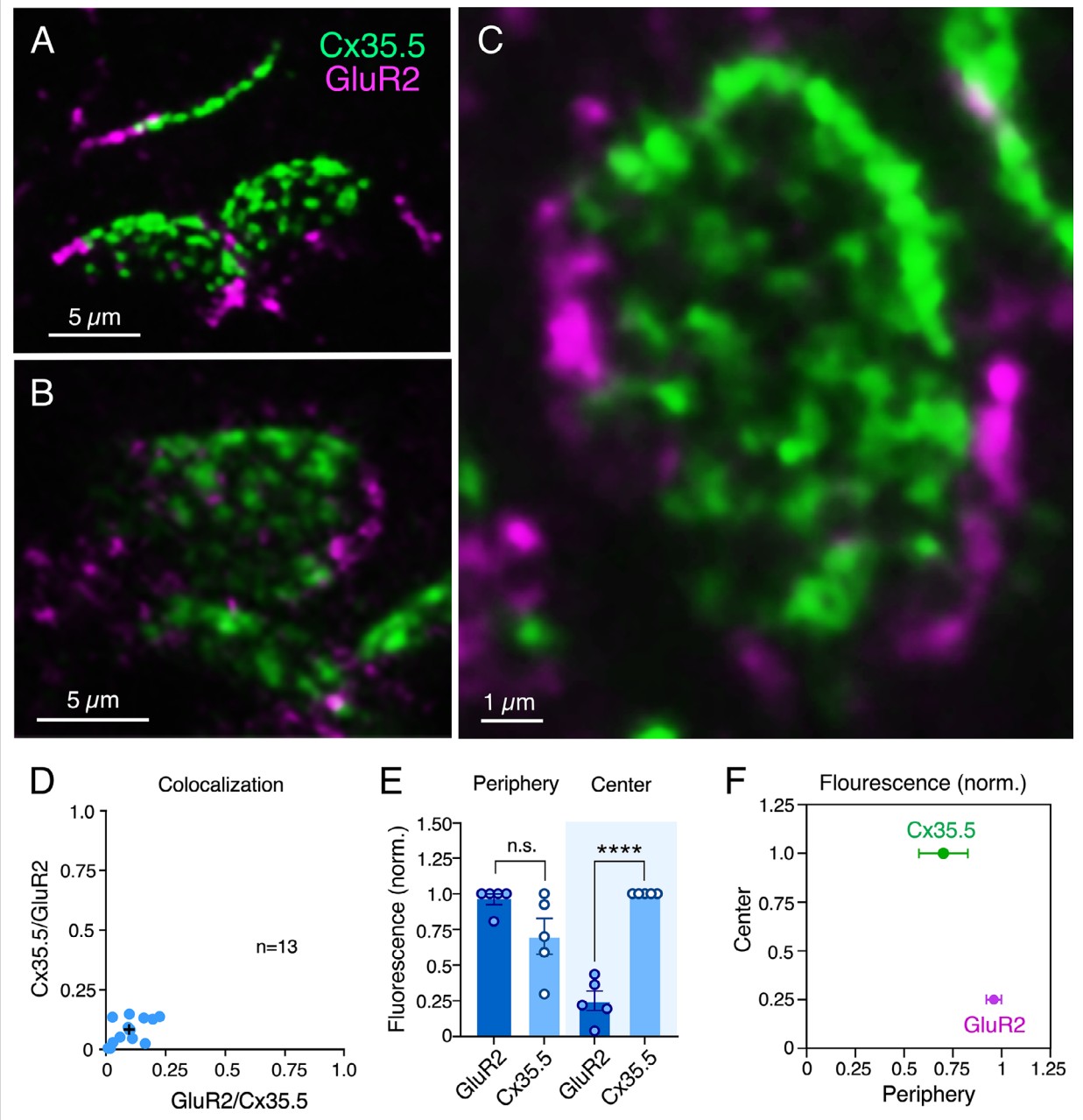

**Figure 2.** Electrical and chemical transmitting areas are mutually exclusive. (**A, B**) Expanded synaptic contact areas labeled with anti-Cx35.5 (green) and anti-GluR2 (magenta) (**A**: projection of 18 sections at 0.55 µm z-step size; **B**: projection of 15 sections at 0.50 µm z-step size). (**C**) 'En face' view of an expanded synaptic contact area showing that GluR2 labeling is restricted to the periphery of the contact, whereas Cx35.5 labeling is distributed throughout the whole contact area (projection of 46 sections at 0.65 µm z-step size). (**D**) Graph shows the lack of colocalization (see 'Materials and methods') between Cx35.5 and GluR2 fluorescence at individual club ending (CE) contacts, determined by the Manders' colocalization coefficient: GluR2/Cx35.5 0.10 ± 0.020 (x-axis); Cx35.5/GluR2 0.08 ± 0.015 (y-axis), n = 13 CEs from five fish. Cross mark indicates the average value. (**E**) Quantification of fluorescence over area for Cx35.5 and GluR2 at the 'Center" (central ¾) and 'Periphery' (remaining ¼) of the CE contact area. Values of fluorescence/area are represented as normalized to those of Cx35.5 in the center (higher value): GluR2 center: 0.25 ± 0.069; Cx35.5 periphery: 0.70 ± 0.126; GluR2 periphery: 0.96 ± 0.039 (n = 5 CEs from five fish). While fluorescence for Cx35.5 and GluR2 is not significantly different in the periphery (n.s.), Cx35.5 distinctly predominates over GluR2 at the center (Student's t-test, p<0.0001). (**F**) Graphical description of the center vs. periphery distribution of Cx35.5 and GluR2 for the data described in (**E**). The scale bars represent actual dimensions; expanded images were not adjusted for expansion factor.

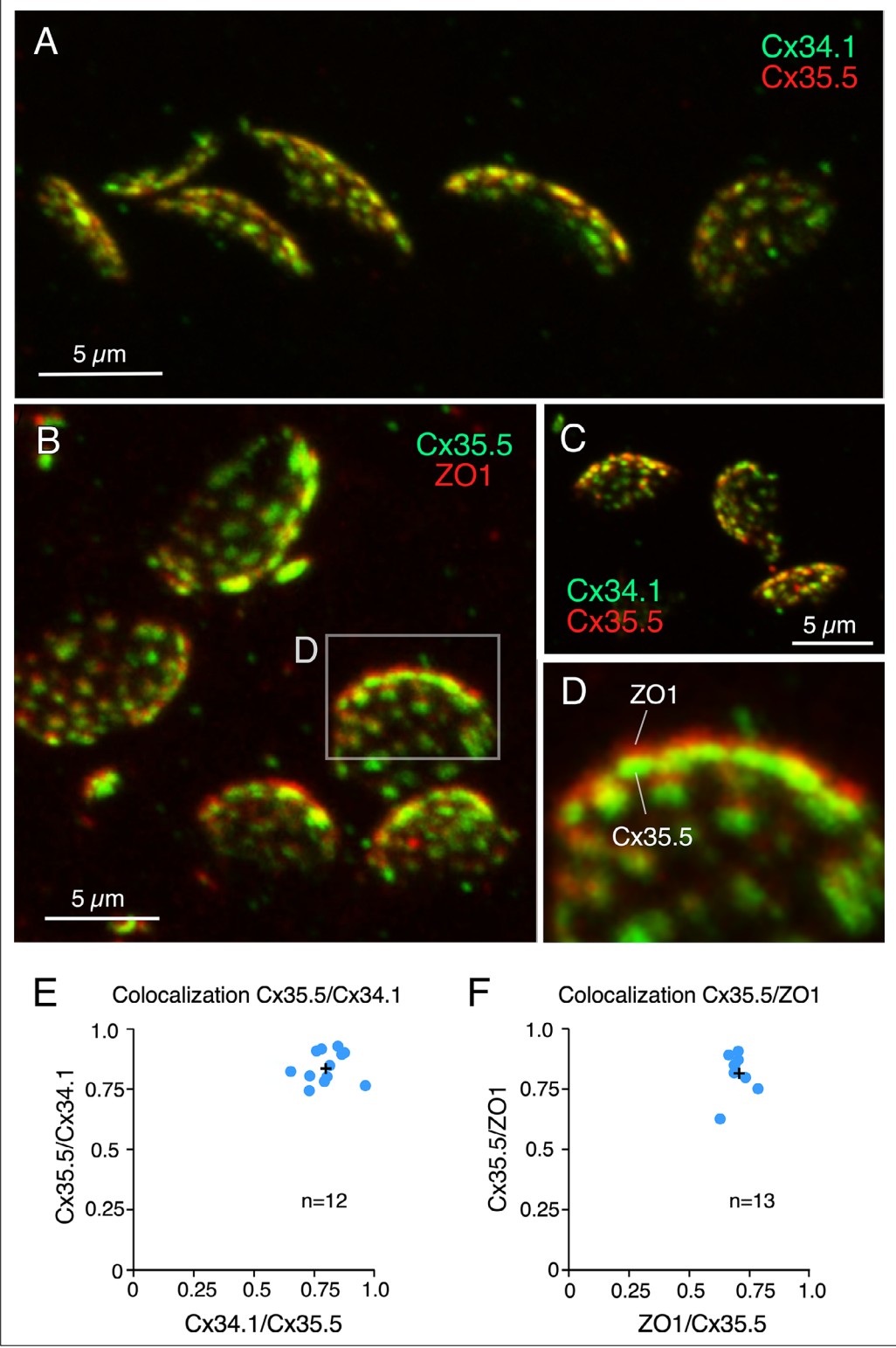

**Figure 3.** Labeling for gap junction proteins reveals the presence of multiple puncta at expanded club ending (CE) synaptic contact areas. (**A**) CE synaptic contact areas labeled with anti-Cx34.1 and anti-Cx35.5 (projection of 69 sections at 0.65 μm z-step size). (**B**) Contact areas labeled with anti-Cx35.5 and anti-ZO1. Same experiment as *Figure 1E* (projection of 19 sections at 0.88 μm z-step size). (**C**) Labeling with anti-Cx34.1 and anti-Cx35.5 (projection of 50 sections at 0.40 μm z-step size). (**D**) Magnification of the boxed region in (**B**) showing a side view of an expanded synaptic contact area labeled for Cx35.5 and ZO1. (**E**) Graph showing colocalization of Cx35.5 and

*Figure 3 continued on next page*

*Figure 3 continued*

Cx34.1 fluorescence at individual CEs determined by the Manders' coefficient: Cx34.1/Cx35.5 0.80 ± 0.023 (x-axis); Cx35.5/Cx34.1 0.84 ± 0.019 (y-axis), n = 12 CEs from six fish. (**F**) Colocalization of Cx35.5 and ZO1 fluorescence at individual CEs. Manders' coefficient: ZO1/Cx35.5 0.71 ± 0.012 (x-axis); Cx35.5/ZO1 0.82 ± 0.021 (y-axis), n = 13 CEs from seven fish. Cross mark indicates the average value. The scale bars represent actual dimensions; expanded images were not adjusted for expansion factor.

The online version of this article includes the following video for figure 3:

**Figure 3—video 1.** Expansion microscopy with anti-Cx35.5 and anti-ZO1.
https://elifesciences.org/articles/91931/figures#fig3video1

proteins (*Figure 4A*) was easier to establish and accurately measure at the periphery of the labeled areas (*Figure 4B*). Line scan of samples labeled for Cx35.5 and Cx34.1 indicated the presence of labeling for these proteins at single punctum (*Figure 4C*). Strikingly, while the peak of maximum intensity for the pre-and postsynaptic connexins was aligned (*Figure 4C*), the peak of maximum intensity for Cx35.5 always preceded that of ZO1 (*Figure 4D*). This is consistent with the fact that, while Cx35.5 is presynaptically localized, ZO1b, one of the two zebrafish orthologs of ZO1, was reported to be post-synaptic. The observed distance between the peaks of fluorescence was not due to the differences in the wavelength of the fluorophores due to uncompensated chromatic aberration as it remained when secondary antibodies were swapped (*Figure 4D and E*). These differences are quantified in the graph of *Figure 4F* (see figure legend for statistical analysis). The finding confirms previous conclusions reached with biochemical and chimera analysis indicating that ZO1b is only located at postsynaptic hemiplaques (*Lasseigne et al., 2021*). Although a distance between Cx35.5 and Cx34.1 labeling peaks was observed in some samples (~0.15 µm), it was still too small to be consistently detected with this method due to the spatial amplification produced by the fluorophores. By increasing the distance between pre- and postsynaptic sites, expansion increases resolution, making it possible to explore differences in the composition of GJ hemiplaques. Thus, altogether, our findings indicate that each puncta correlates to a GJ.

Puncta labeled for Cx35.5 (*Figure 5A and B*), which we established each represent a GJ, showed a high degree of variability in number and size (see inset of *Figure 5C*). We then quantified the number of puncta and their individual area in 'en face' views of CEs, at which this analysis was possible. (Panels B and C in *Figure 5* are the same experiment as *Figures 1E and 3B and C*, illustrating the ability of expansion microscopy for providing multiple layers of information within the same experiment.) *Figure 5D* illustrates the variability in puncta area observed between neighboring CEs within the same M-cell lateral dendrite in three different fish. Histograms show a similar variability in number and size at all terminals. Overall, the number of GJs per CE averaged 36.73 ± 1.287 (n = 11 CEs from four fish), and their area ranged from 0.06 to 1.99 µm². The number of GJs might have been slightly underestimated because fluorophore spatial amplification could have caused a big punctum to form from two closely spaced small GJs. This limitation could result in two nearby GJs appearing to merge. After correcting for the expansion factor (13×), the areas of the GJs were estimated to be between 4 and 148 nm². Assuming that connexons in GJ plaques are organized in a crystalline fashion with a density of 12,000 connexons/µm² (*Kamasawa et al., 2006*), we estimated that GJs at CEs contain 49–1775 connexons, and the total junctional area represents an average of 12,425 ± 493.76 connexons per CE. Thus, electrical transmission at zebrafish CEs is mediated by multiple GJs containing a variable number of channels.

Our labeling with Cx35.5 and GluR2 showed that most of the contact area of a CE operates as an electrical synapse. We then asked what other associated anatomical structures might be contributing to electrical transmission. Previous electron microscopy analysis exposed the presence of AJs in close proximity to neuronal GJs, including those at CEs in goldfish and larval zebrafish (*Figure 6A*). AJs are known to initiate and mediate the maturation and maintenance of cell–cell contacts, including GJs, at which a wealth of evidence suggests a close functional interaction (*Shaw et al., 2007*; *Defourny and Thiry, 2021*; *Thomas et al., 2021*). Therefore, we decided to investigate the incidence, association, and spatial distribution between these two structures at CEs. Recent immunohistochemical analysis revealed the association between components of AJs and Cx36 in various mammalian structures (*Nagy and Lynn, 2018*). Expansion for Cx35.5, transmembrane protein N-cadherin, and the intracellular protein ß-catenin, the latter two being major structural components of AJs, showed that

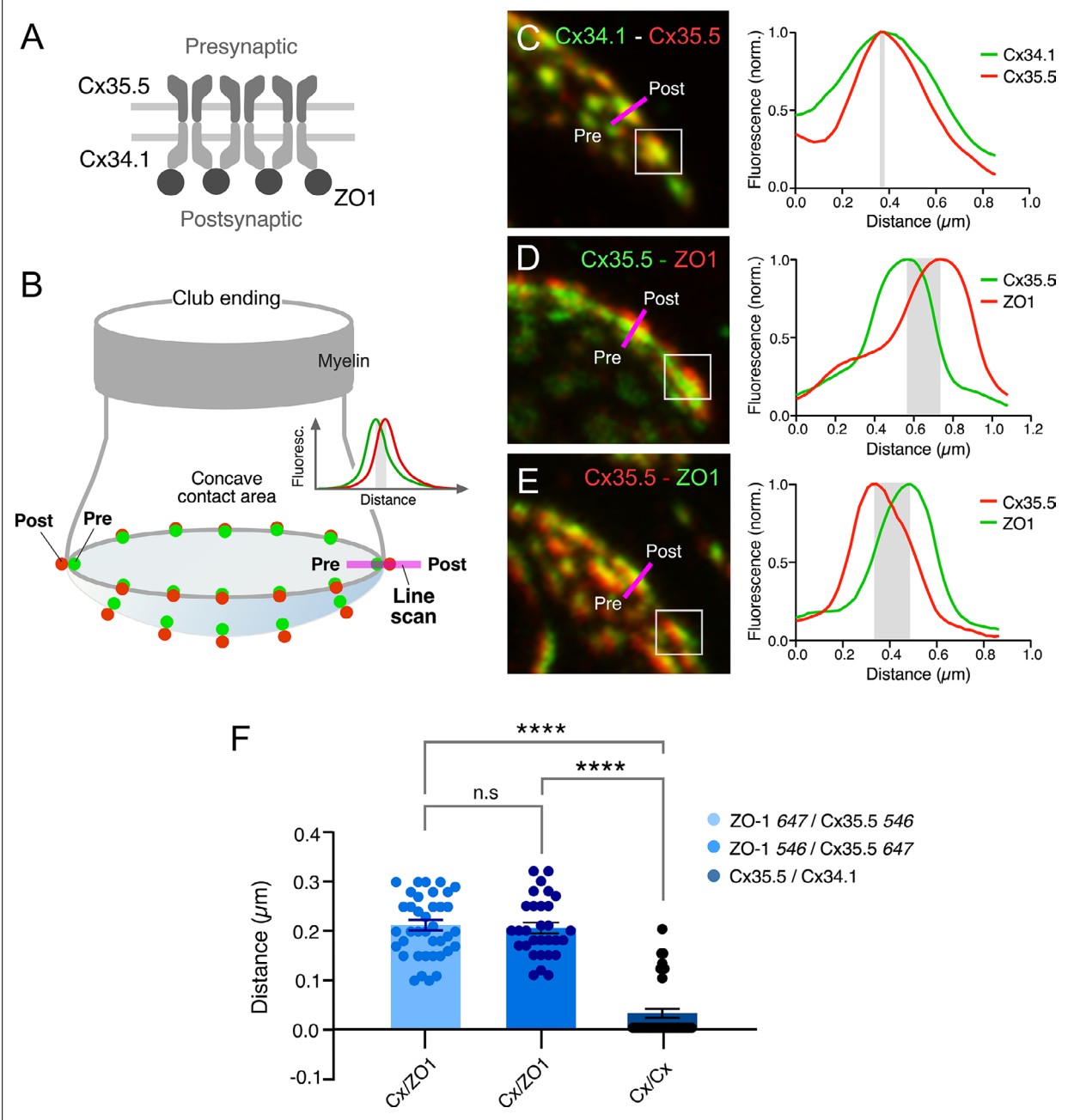

**Figure 4.** Expansion microscopy reveals the molecular components of gap junction (GJ) plaques at club ending (CE) synaptic contact areas.
(**A**) Schematic representation of the molecular organization of GJs between CEs (presynaptic) and the M-cell (postsynaptic). The presynaptic and postsynaptic hemichannels are formed by Cx35.5 and Cx34.1, respectively. The scaffolding protein, ZO1, is postsynaptic and interacts with Cx34.1. (**B**) Cartoon of a CE terminal illustrating the concavity of its contact area with the M-cell. The concavity determines differences in the relative position of presynaptic (green) vs. postsynaptic (red) labeling at different points throughout the contact area. Puncta located in the periphery of the contact are ideally aligned to determine colocalization of fluorescence at individual puncta (line scan, inset). (**C–E**) Line scan of puncta at expanded contact areas showing colocalization of presynaptic Cx35.5 and postsynaptic Cx34.1 (**C**) (projection of 69 sections at 0.65 μm z-step size), and presynaptic Cx35.5 and postsynaptic ZO1 (**D–E**) (**D**: projection of 26 sections at 0.86 μm z-step size; **E**: projection of 55 sections at 0.60 μm z-step size). The example in (**C**) is part of the experiment illustrated in *Figure 3A*. The magenta lines indicate the position of the line scan in each case. The fluorescence intensity profiles for each fluorophore are illustrated on the right side of each panel. As a control, secondary antibodies were swapped in (**E**). (**F**) Bar graph illustrates the distance between the peaks of fluorescence intensity profiles for Cx35.5-Cx34.1 labeling (with either 647Atto or 546Alexa-Cx35.5 vs. either 647Atto or 546Alexa-Cx34.1: 0.03 ± 0.010 μm, n = 37 puncta from six fish) and Cx35.5-ZO1 labeling (546Alexa-ZO1 vs. 647Atto-Cx35.5: 0.21 ± 0.011 μm, n = 30 puncta from three fish). Secondary antibodies were swapped as control (647Atto-ZO1 vs. 546Alexa-Cx35.5: 0.21 ± 0.011 μm, n = 39 puncta from eight fish). Bars represent ± SEM (ANOVA analysis with Tukey's multiple comparison test correction; ****p<0.0001). The scale bars represent actual dimensions; expanded images were not adjusted for expansion factor.

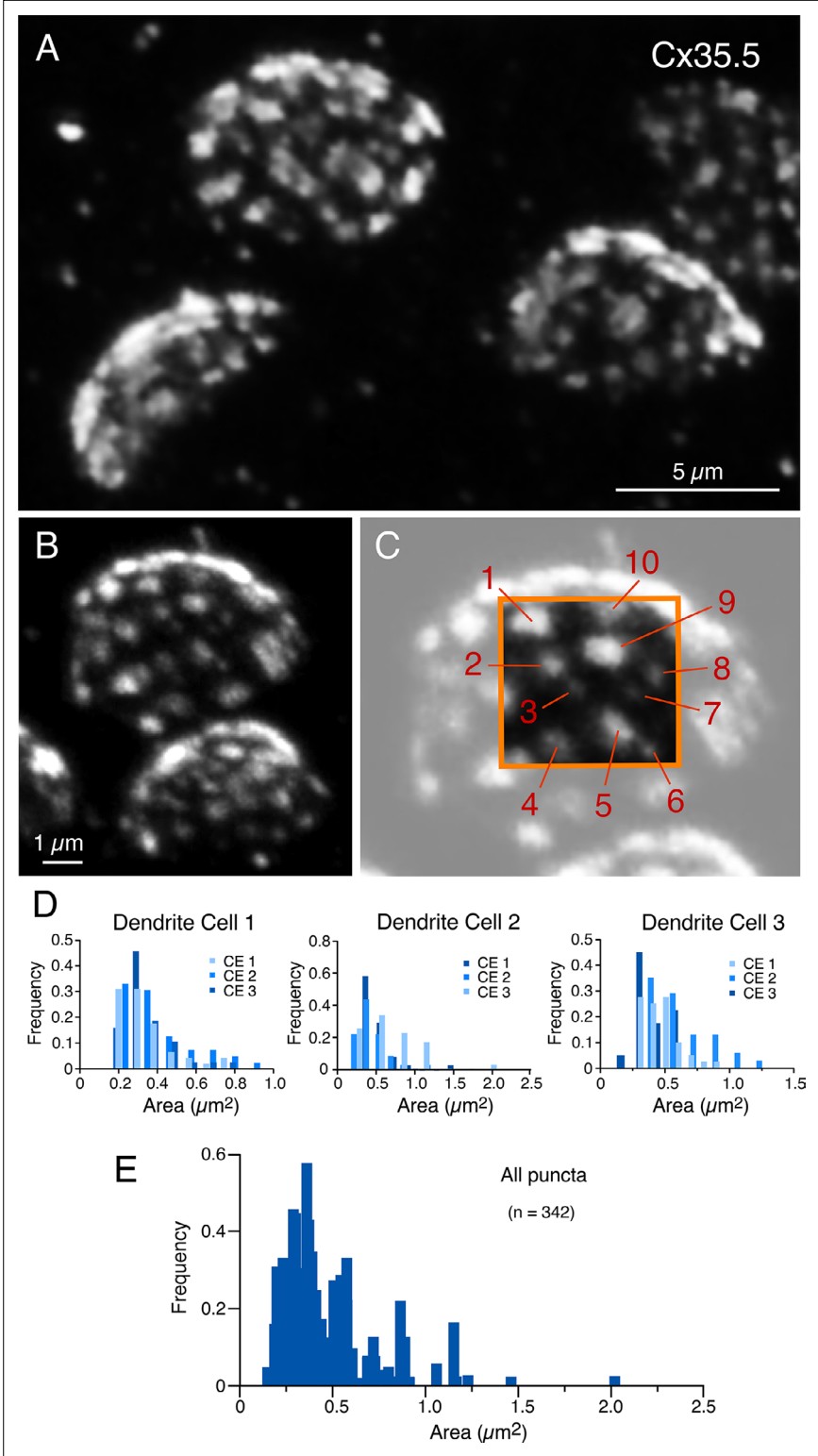

**Figure 5.** Expansion reveals the presence of multiple, variably sized, gap junctions. (**A, B**) 'En face' views of expanded club ending (CE) contact areas labeled with anti-Cx35.5 showing multiple puncta with high variability of their size (**A**: projection of 18 sections at 0.60 μm z-step size; **B**: projection of 19 sections at 0.88 μm z-step size). (**C**) Magnification of the CE contact area at the top of (**B**) (light gray). The area enclosed by the orange box illustrates the wide variability in puncta size, labeled 1–10 (to better highlight the variability in puncta size, the image delimited by the orange box was cropped from and placed on the same region of the lighter image). Panels

*Figure 5 continued on next page*

*Figure 5 continued*

(**B**) and (**C**) are the same experiment as *Figures 1E and 3B–D*, demonstrating the ability of expansion microscopy for providing multiple layers of information within the same experiment. (**D**) Frequency histograms summarize the number and size distribution of puncta labeled for Cx35.5 obtained from three dendrites, each belonging to a different fish (bar graphs labeled as 'Dendrite cell 1', 'Dendrite cell 2', 'Dendrite cell 3'). Each histogram illustrates, overlapped in different shades of blue, the values obtained from the analysis of three 'en face' views of CE terminals. Histograms show similar variability in number and size for all nine terminals. (**E**) Frequency histogram summarizes the average values resulting from the analysis of all nine 'en face' CE views. The scale bars represent actual dimensions; expanded images were not adjusted for the expansion factor.

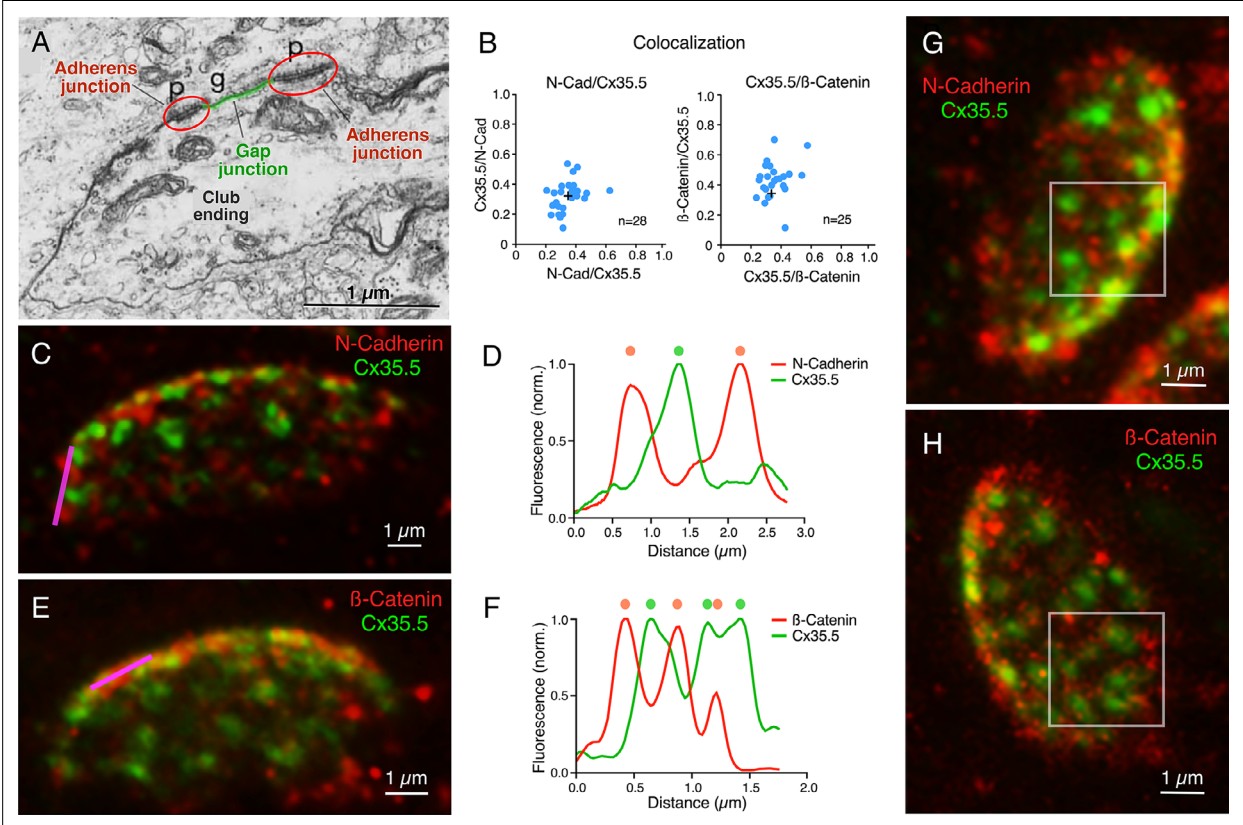

**Figure 6.** Gap junctions (GJs) at club endings (CEs) are associated with adherens junctions (AJs). (**A**) Electron micrograph of a CE obtained in a 6 days post fertilization (dpf) zebrafish showing a GJ (g, highlighted in green) surrounded by AJs (p, encircled in red). ((**A**) was reproduced from Figure 5 of *Kimmel et al., 1981*, with permission from John Wiley and Sons. It is not covered by the CC-BY 4.0 license and further reproduction of this panel would need permission from the copyright holder; *Kimmel et al., 1981*.) (**B**) GJ and AJ proteins do not colocalize. Left: Cx35.5 and N-cadherin labeling show a low index of colocalization. Manders' coefficient: N-cadherin/Cx35.5 0.35 ± 0.016 (x-axis); Cx35.5/N-cadherin 0.32 ± 0.019 (y-axis), n = 28 CEs from 10 fish. Right: Cx35.5 and β-catenin labeling also show low colocalization. Manders' coefficient: Cx35.5/β-catenin 0.36 ± 0.017 (x-axis); β-catenin/Cx35.5 0.43 ± 0.024 (y-axis), n = 25 CEs from 12 fish. Cross mark indicates the average value. (**C**) Expansion microscopy of a CE contact area labeled for N-cadherin (red) and Cx35.5 (green) (projection of 47 sections at 0.40 μm z-step size). (**D**) Fluorescence profiles for N-cadherin and Cx35.5 obtained with a line scan (magenta line in **C**) are mutually exclusive. The small degree of colocalization observed in (**B**) is likely due to fluorophore amplification and the close spatial association between GJs and AJs, as shown in (**A**). (**E**) Image shows an expanded CE contact area labeled for β-catenin (red) and Cx35.5 (green) (projection of 12 sections at 0.60 μm z-step size). (**F**) Line scan (magenta line in **E**) shows that labeling for β-catenin and Cx35.5 is also mutually exclusive. (**G, H**) 'En face' view of the expanded contact area double-labeled for N-cadherin and Cx35.5, and β-catenin and Cx35.5, respectively, showing the close association of GJs and AJs throughout the synaptic contact area. Insets: the boxed areas in the 'en face' images highlight the mutually exclusive labeling (**G**: projection of 24 sections at 0.50 μm z-step size; **H**: projection of 16 sections at 0.60 μm z-step size). The scale bars represent actual dimensions; expanded images were not adjusted for the expansion factor.

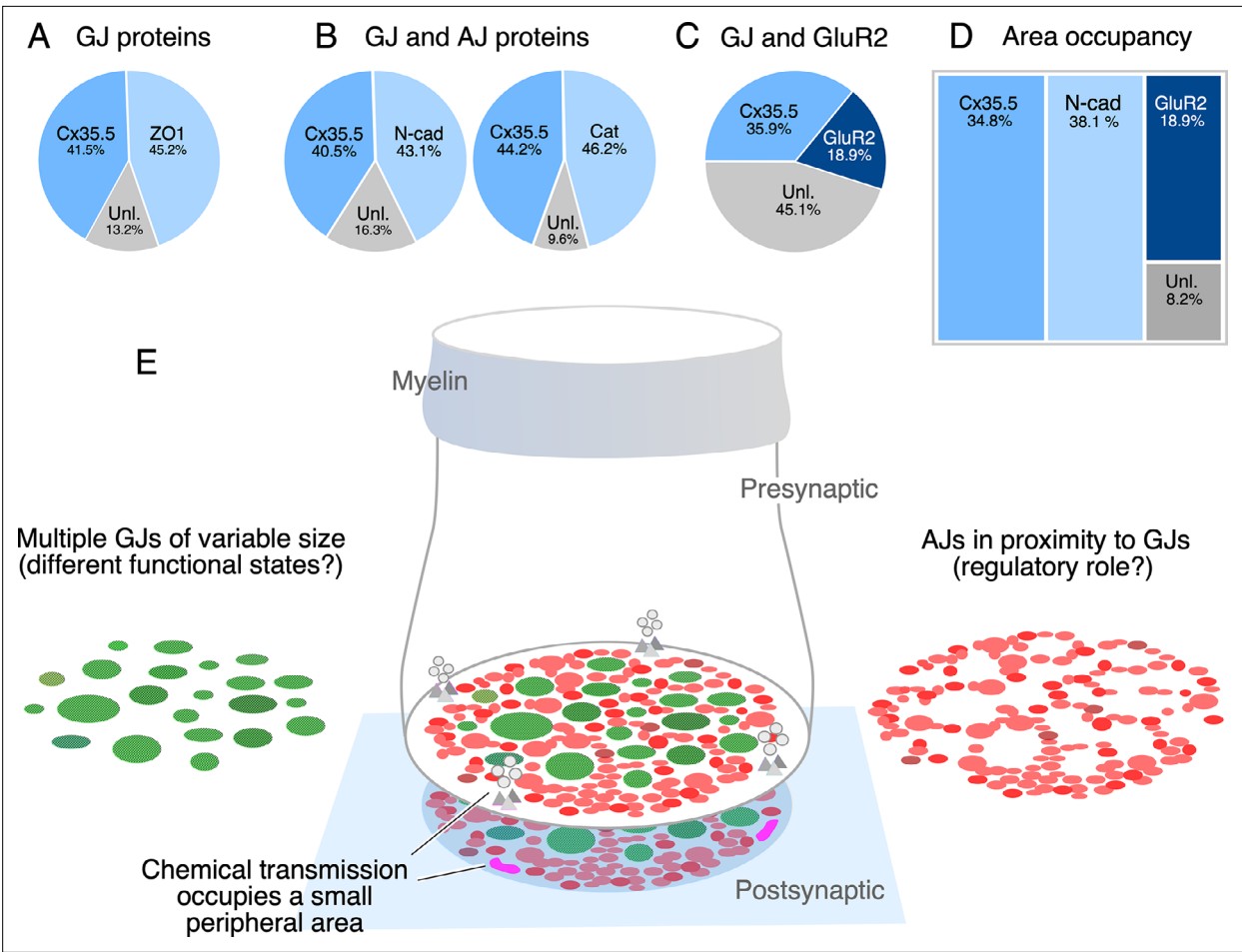

**Figure 7.** The electrical synapse at the club ending (CE) combines multiple gap junctions (GJs) with adherens junctions (AJs). (**A**) Double labeling with anti-Cx35.5 and anti-ZO1 shows a similar proportion of fluorescence at CEs (ZO1 45.24% ± 2.709; Cx35.5 41.53% ± 4.058; unlabeled 13.23%; n = 8 CEs from six fish). The region of interest (ROI) for analysis of fluorescence here and (**B**) was defined by the outline of Cx35.5 labeling of CEs. (**B**) Double labeling for N-cadherin and Cx35.5 (left), and for β-catenin and Cx35 (right) also shows similar proportionality (N-Cad 43.20% ± 3.334; Cx35.5 40.48% ± 3.041; unlabeled 16.32%; n = 13 from five fish; β-catenin 46.21% ± 2.728; Cx35.5 44.18% ± 1.671; unlabeled 9.61%; n = 13 CEs from seven fish). (**C**) Double labeling for Cx35.5 and GluR2 shows lack of proportionality, with Cx35.5 fluorescence occupying the majority of the CE contact area (GluR2 18.99% ± 1.601; Cx35.5 35.92% ± 2.087; unlabeled 45.10%; n = 10 CEs from five fish). ROI was defined in this case by the outline of GluR2 labeling. (**D**) Tree plot illustrating the area occupancy (fluorescence/contact area) for AJ (N-Cad = 38.1%), GJ (Cx35 = 34.8%), and glutamatergic (GluR2=18.9%) labeling at individual CEs (values normalized to ROI outlined by GluR2 labeling). The unlabeled area represents 8.1% of the contact's surface. (**E**) The cartoon summarizes the synaptic components identified at a single CE contact. While chemical synapses are restricted to a small and peripheral area of the contact (presynaptic vesicles and release sites are represented in gray, postsynaptic receptor areas in magenta), most of its contact surface is occupied by multiple GJs (green) of variable size, which are interleaved and closely associated to AJs (red).

labeling for AJs components seems equally distributed through the synaptic contact (*Figure 6B–D*). In contrast to Cx35.5, which is characterized by well-defined puncta reminiscent of GJs, labeling for N-cadherin and ß-catenin was diffuse, less structured, and did not colocalize with Cx35.5 (*Figure 6B*; see figure legend for statistical analysis). Rather, the labeling for N-cadherin (*Figure 6C*) and ß-catenin (*Figure 6E*) was interleaved with Cx35.5 puncta, as illustrated by line scan analysis in the margins of CE contacts (*Figure 6D and F*). Furthermore, in 'en face' views, labeling for N-cadherin (*Figure 6G*) and ß-catenin (*Figure 6H*) was broadly distributed throughout the surface of the contact, appearing to engulf Cx35.5-labeled puncta. While labeling for N-cadherin and ß-catenin was interleaved with Cx35.5 labeling, we observed some degree of colocalization (*Figure 6B*) likely due to fluorophore amplification of the close spatial association between GJs and AJs (see *Figure 6A*).

The spatial distribution of the labeling for components of AJs suggests that their function is associated with electrical transmission, rather than chemical transmission. To establish this association, we measured the relative proportion of labeling fluorescence of various synaptic components at CE

contact areas. As expected, labeling for Cx35.5 and the GJ scaffold ZO1 was equally proportional (*Figure 7A*). Similar proportionality was found for fluorescence of N-cadherin and Cx35.5, and ß-catenin and Cx35.5 (*Figure 7B*). In contrast, fluorescence of GluR2 represented a smaller fraction than that of Cx35.5 labeling (*Figure 7C*; see figure legend for statistical analysis). Finally, we normalized the fluorescence to the area of the CE contact to calculate the area occupancy for these three synaptic structures (*Figure 7D*). Less than 19% was occupied by GluR2, while around 73% of the contact was occupied by similar amounts of Cx35.5 and AJs. About 8% of the contact area was not labeled for these antibodies, either corresponding to the surface membrane between the structures recognized by labeling or to the presence of additional synaptic components. Thus, while chemical synapses are restricted to a small and peripheral area of the contact, most of the contact surface is occupied by multiple GJs of variable size which are interleaved and closely associated to AJs (*Figure 7E*).

## Discussion

"One of the most difficult problems in the correlation of structural and functional concepts is that of accurately defining the synapse" (*Robertson et al., 1963*).

Because they support both electrical and chemical synaptic transmission, CEs on the M-cells enabled the correlation of their synaptic properties with structural specializations for each of these modalities of communication (*Robertson et al., 1963*). In addition to specializations observed at purely chemically transmitting contacts, electron microscopy analysis by *Robertson, 1963* revealed areas of close membrane apposition that at 'en face' views were round-shaped and exhibited a characteristic reticular pattern. These 'synaptic discs' provided early evidence for the cellular structures that we now know as 'gap junctions' (GJs): clusters of intercellular channels that provide the mechanism of communication for electrical transmission. However, in his seminal paper (*Robertson et al., 1963*), Robertson warned about the limitations of correlating structural and functional notions to define the components of an electrical synapse. In contrast to chemical synapses, it is challenging to define exactly what anatomically constitutes an electrical synapse as neuronal GJs are often found connecting cell somata or other neuronal processes, such as dendrites and axons (*Nagy et al., 2018*). Because the presynaptic bouton anatomically marks the limit of a synapse, CEs provided the ideal opportunity to investigate the components of an electrical synapse. By applying expansion microscopy to these single terminals, we found evidence suggesting that, in contrast to the general perception, electrical synapses might have a more complex structural organization. Our data indicates that the CE electrical synapse operates with not one but multiple (~35) GJs that are in close association with structural and signaling molecules known to be components of AJs (*Figure 7E*). This extended notion of the overarching organization of an electrical synapse might contribute to a better understanding of their functional diversity and structural configurations.

### Expansion microscopy of a single synaptic contact

Because of their identifiability, CEs have historically been amenable for exploring synaptic structure and function with novel technical approaches. Here we applied expansion microscopy to these terminals, which allowed us to generate a 'map' of the distribution of its various synaptic proteins. We found that most of the contact area of a CE operates as an electrical synapse and only a peripheral ~19% is dedicated to chemical transmission. The 4× diameter increase after expansion represented a 13× increase of the oval contact areas and did not alter the morphology of the terminals, as supported by (1) expansion reproduced the characteristic concavity of the CE synaptic contact area with the M-cell; (2) the spatial distribution of chemical vs. electrical synaptic areas was consistent with that observed by electron microscopy in adult goldfish (*Tuttle et al., 1986*) and unexpanded zebrafish terminals *Lasseigne et al., 2021*; (3) labeled puncta contain all the molecular components known to form GJs at CEs; (4) morphometric analysis indicated that the expansion process had no selective effects across the CE population; and (5) finally, consistent with this analysis, tissue expansion was tridimensional and exposed the expected pre- vs. postsynaptic localization of proteins at single punctum representing individual GJs.

The properties of any 'map' are determined by the methodology that was employed to create it. While the sentence "The map is not the territory" was coined by Alfred Korzybski to metaphorically illustrate the distinction between brain perception and reality *Korzybski, 1933*, it applies to the

properties of maps in general and to the usefulness of applying expansion microscopy to expose the synaptic map of CE contacts. Moreover, any useful map should not necessarily be fully accurate (discussed in "Of exactitude in science" by *Borges, 1998*), but capable of capturing an element of reality. As a representation of reality, the labeling of molecules forming GJs, glutamate receptors, and AJs allowed us to expose the incidence and spatial distribution of the various synaptic components with enough accuracy to provide an 'analogous structure' of the contact's structural organization. Expansion microscopy is complementary to a different representation of reality, electron microscopy, which is capable of providing fine details of the structures, but can inform, to a lesser degree, their biochemical composition. Unlike the more labor-intensive electron microscopy, expansion microscopy was more easily capable of exposing the complete contact area of the terminal, while additionally revealing its biochemical composition (i.e., because of its bias toward exposing intramembrane particles, freeze fracture EM of CEs did not reveal additional synaptic components between GJs; *Yao et al., 2014*; *Tuttle et al., 1986*; *Kohno and Noguchi, 1986*; *Pereda et al., 2003*). Thus, without having its structural resolution, expansion microscopy was able to expose, with sufficient resolution, a complementary version of spatial features that were only observed so far with electron microscopy.

Furthermore, because of its tridimensional nature, expansion microscopy not only enlarged the synaptic contact area, but also the distance between synaptic elements, particularly enhancing the detection of pre- vs. postsynaptic components at GJs. Ultrastructural images and the bilateral requirement of GJ hemichannels convey the perception that GJs are symmetrically organized structures. Emerging evidence obtained at CEs suggests otherwise. Recent data indicates that GJ asymmetry at CEs is not only restricted to the connexin composition of pre- and postsynaptic hemichannels, but also to scaffolding molecules (*Lasseigne et al., 2021*; *Martin et al., 2023*; *Miller et al., 2017*; *Rash et al., 2013*; *Miller et al., 2015*). That is, biochemical data and chimera analysis suggested that ZO1b, one of the two zebrafish orthologs of ZO1 whose function is critical for the presence of connexins at electrical synapses, is located in the postsynaptic hemiplaque from where it exerts its critical function (*Lasseigne et al., 2021*). Confirming this conclusion, our expansion microscopy unambiguously exposed the postsynaptic location of ZO1b. Moreover, because the antibody recognizes both ZO1b and ZO1a, our data indicates that, although with different functions, both ZO1 orthologs might be restricted to postsynaptic hemiplaques. Together with the genetic accessibility of zebrafish, the synaptic map of CEs provided by expansion microscopy will facilitate exploring the functional relationship between the structures supporting electrical transmission and its regulation at each side of the junction.

## Functional association between gap junction and adherens junctions

Electron microscopy of CEs in both adult goldfish and larval zebrafish revealed the presence of AJs in the vicinity of GJs (*Kohno and Noguchi, 1986*; *Kimmel et al., 1981*), an association which was also observed at neuronal GJs in viper cerebellum (*Sotelo, 1977*) and cat inferior olive (*Sotelo et al., 1974*). AJs are dynamic cell–cell adhesion complexes that are continuously assembled and disassembled. They are formed by a complex of proteins that include cadherins and catenins (ß and α). Growing evidence indicates a functional relationship between AJs and GJs (*Shaw et al., 2007*; *Meyer et al., 1992*; *Li et al., 2005*; *Wei et al., 2005*; *Dalva et al., 2007*). While it was generally accepted that GJs form via lateral diffusion of hemichannels following microtubule-mediated delivery to the plasma membrane, more recent evidence obtained in cell expression systems shows that microtubules actually tether to the AJ, facilitating delivery of vesicles containing connexin hemichannels directly to the cell–cell border of the GJ (*Shaw et al., 2007*). This process depends on the interaction of microtubules with plus-end-tracking protein (+TIP) EB1, its interacting protein p150 (Glued), and the AJ proteins N-cadherin and ß-catenin (*Shaw et al., 2007*). Furthermore, the evidence indicates that this process also requires homophilic interactions between N-cadherins. Similar peripheral delivery of GJ channels to AJs was observed in sensory epithelial cells of the cochlea (*Defourny and Thiry, 2021*), indicating that this mechanism also operates in tissues. Also consistent with a close functional relationship between these structures, AJ formation hierarchically regulates the formation of GJs in cardiac pacemaker cells (*Thomas et al., 2021*), adjusting the excitability and coupling of these neurons in the context of their pacemaking function.

## The electrical synapse at CEs operates with multiple GJs

The investigation of the mechanisms and structures underlying electrical transmission has generally centered on the properties of GJ channels and their supporting molecules, a perspective that is restricted to a mechanisms occurring at a single GJ plaque. However, electrical transmission at each zebrafish CE terminal is mediated on average by about 35 GJs, suggesting that they together operate as a functional unit. Like chemical synapses that operate with dramatically different numbers of releases sites, ranging from a single one at synaptic terminals of PHP cells on the M-cell (*Korn et al., 1982*; *Korn et al., 1981*) to up to 700 in the Calyx of Held (*Sätzler et al., 2002*; *Taschenberger et al., 2002*), electrical synapses could function with different number of GJs. Operating with multiple GJs might result in more reliable electrical transmission as only a very small fraction of channels are known to be functional at individual GJs (*Flores et al., 2012*; *Curti et al., 2012*; *Szoboszlay et al., 2016*; *Marandykina et al., 2013*). Because of the wide variation in size, GJs at CEs might coexist at different states of conductance (*Bukauskas et al., 2000*). The presence of multiple GJs of different sizes might be reflective of high plastic regulation, a possibility consistent with the dynamic properties of CE electrical synapses, which are known to undergo activity-dependent potentiation of their synaptic strength (*Yang et al., 1990*; *Pereda and Faber, 1996*; *Pereda et al., 1998*; *Cachope et al., 2007*).

While it is currently unknown how regulation of the overall conductance of the CE electrical synapse is achieved, it must necessarily result from the coordinated contribution of its multiple GJs, suggesting the need of additional synaptic structures capable of coordinating the function of the numerous GJs. Given the close spatial association that we report here, it is tempting to speculate that interactions between AJs and GJs could underlie such function. Since interactions between GJs and AJs involving cadherins are thought to be relevant for the insertion of new GJ channels (*Defourny and Thiry, 2021*; *Shaw et al., 2007*), this process might be intimately related to the maintenance and plasticity of GJ communication via regulated turnover of these channels at CEs, which are formed by fish homologs of Cx36 (*Rash et al., 2013*; *Miller and Pereda, 2017*). Consistent with the possibility of AJs promoting the insertion of new GJ channels at CEs, microfilaments arriving to an AJ situated in close proximity to a GJ can be observed in an image of the EM study by Kohno and Naguchi (see Figure 2 in *Kohno and Noguchi, 1986*). Operating with multiple GJs also provides electrical synapses with the possibility of either silencing or activating individual GJs as a potential additional mechanism of strength regulation. Finally, although N-cadherin was also shown to be involved in regulating chemical synapses (*Korn et al., 1981*), the distribution of chemical synapses to small peripheral areas of the CE contact indicates a primary functional role of this molecule at electrical synapses as both GJ proteins and N-cadherin are similarly distributed throughout the entire synaptic contact area. Future investigations on the functional association between GJs and AJs will shed light on the functional organization of electrical synapses.

## The components of the electrical synapse

The structural complexity of chemical synapses has long been recognized. Structural complexity is also a hallmark of immunological synapses (*Dustin, 2014*), specialized functional contacts which quickly assemble between a thymus-derived lymphocyte and an antigen-presenting cell (*Dustin, 2012*; *Dustin and Groves, 2012*). Both chemical and immunological synapses share a general molecular organization combining cell adhesion molecules, usually restricted to the periphery of the contacts, with those more centrally located and responsible for providing intercellular communication (*Dustin and Colman, 2002*). Our results indicate that this structural arrangement might also apply to the CE electrical synapse. That is, our finding that AJs surround GJs all through the contact is consistent with the general organization of chemical and immunological synapses, at which adhesion molecules are located in the periphery of the communicating mechanism. Thus, just as chemical and immunological synapses, electrical synapses also seem to combine cell adhesion molecules with those responsible for mediating intercellular communication. Based on these similarities in the overall organization across synapses and evidence of functional interactions with GJs (see above), we propose that AJs could be considered components of electrical synapses. While additional components are likely to contribute to electrical transmission, the distribution of adhesion molecules at electrical synapses might serve to define their synaptic boundary, an arrangement that might be used as a template to identify electrical synapses in other structures. Supporting this notion, a close association between components of

AJs and Cx36 was observed at various mammalian brain structures (*Nagy and Lynn, 2018*) and was suggested they share a common molecular complex (*Nagy and Lynn, 2018*). As in CEs, multiple Cx36-labeled puncta engulfed by N-cadherin labeling were also observed at cell–cell contacts between the somata of neurons of the mesencephalic nucleus of the trigeminus in mice (see Figure 8 in *Nagy and Lynn, 2018*), indicating that this arrangement is not a unique feature of fish mixed synapses but might also apply to electrical synapses elsewhere.

In summary, as single synaptic contact, expansion microscopy of CEs offered the possibility of identifying the structures that support electrical transmission and, therefore, the components that together form an electrical synapse, which can be used to define electrical synapses throughout animal connectomes. Moreover, defining the components of an electrical synapse will help expose their functional diversity as different synapses might operate with different numbers of GJs and synaptic arrangements.

## Materials and methods
### Experimental model and subject details
All experiments were performed in 5 dpf zebrafish, *Danio rerio*. Zebrafish were housed at the Department of Neuroscience zebrafish facility, bred and maintained at 28°C on a 14 hr light/10 hr dark cycle. Experiments were carried out in the Tol-056 enhancer trap line (*Satou et al., 2009*).

### Immunohistochemistry
Zebrafish larvae were anesthetized with 0.03% MS-222 (tricaine methanesulfonate) and fixed for 3 hr with 2% trichloroacetic acid, in 1× PBS, at room temperature. Fixed samples were then washed three times with 1× PBS, followed by a brain dissection with the help of a custom-made tungsten needle and forceps. The needle was also used to remove the cerebellum, optic tectum, and telencephalon to accommodate for the working distance of the microscope objective. This step was not necessary in samples used for expansion methods. The dissected brains were washed with 1× PBS + 0.5% Triton X-100 (PBS-Trx) and blocked with 10% normal goat serum + 1% DMSO in PBS-Trx. Brains were then incubated with a primary antibody mix at room temperature overnight. The antibody mix, with block solution, included combinations of the following: rabbit anti-Cx35.5 (*Miller et al., 2015*, clone 12H5, 1:200), mouse IgG1 anti-Cx35/36, which labels both Cx35.5 and Cx35.1 (Millipore, Cat# MAB3045, 1:250), mouse IgG2A anti-Cx34.1 (*Miller et al., 2015*, clone 5C10A, 1:200), mouse IgG1 anti-ZO1 (Invitrogen, Cat# 33-9100, 1:200), mouse IgG1 anti-N-cadherin (BD Transduction Laboratories, Cat# 610920, 1:50), mouse IgG1 anti-beta-catenin (Sigma, Cat# C7207, 1:100), rabbit IgG anti-GluR2/3 (Millipore, Cat# 07-598, 1:200), and chicken IgY anti-GFP (Abcam, Cat# ab13970, 1:200). After three washes in PBS-Trx, brains were blocked with 10% normal goat serum + 1% DMSO in PBS-Trx, followed by incubation in secondary antibody mix at room temperature for 4 hr. Secondary antibody mixes included, in addition to block solution, combinations of mouse IgG- Alexa Fluor 546 (Invitrogen, Cat# A11030, 1:200), mouse IgG- Alexa Fluor 647 (Invitrogen, Cat# A21235, 1:200), rabbit IgG- Alexa Fluor 546 (Invitrogen, Cat# A11010, 1:200), rabbit IgG- Atto 647N (Sigma-Aldrich, Cat# 40839, 1:200), mouse IgG- Atto 647N (Sigma-Aldrich, Cat# 50185, 1:200), and chicken IgY- Alexa Fluor 488 (Invitrogen, Cat# A11039, 1:200). Samples were washed with 1× PBS four times and overnight at 4°C, and then transferred in the dark, onto a slide, and mounted with ProLong Gold antifade (Invitrogen, Cat# P36930). Finally, samples were covered using the 'bridge' procedure (*Martin et al., 2022*) and sealed with nail polish. Samples for the expansion procedure were not mounted as noted here; instead, the samples were incubated in anchoring solution (see below) overnight.

### Expansion
Expansion of brain samples was performed following previous protocols (*Asano et al., 2018*) with additional modifications using the following reagents (final concentrations are reported): anchoring solution: acryloyl-X, SE (0.01%) and PBS (1×) to a complete volume; monomer solution/gelling solution (4-HT, tetramethylethylenediamine [TEMED], and ammonium persulfate (APS) should be added sequentially, one at a time, to each sample): acrylamide (2.5%), N,N' methylenebisacrylamide (0.15%), NaCl (2 M), PBS (1×), sodium acrylate (8.6%), 4-hydroxy-TEMPO (4-HT) (0.01%), TEMED (0.2%), APS (0.2%), and cell culture grade water to a complete volume; digestion buffer: Tris (pH 8.0) (50 mM),

EDTA (1 mM), Triton X-100 (0.5%), NaCl (0.5 M), and cell culture grade water to a complete volume; and proteinase K (8 units/ml). Upon completion of immunolabeling steps, the sample was incubated in anchoring solution overnight. The sample was then washed twice in 1× PBS and placed into monomer/ gelling solution, and placed at 4°C for 50 min, and then at 37°C for 2 hr. Once the hydrogel polymerized, the sample was placed at 50°C for 12 hr in the digestion buffer + proteinase K. Samples were then washed five times with cell culture grade water for expansion. Following this procedure, the hydrogels were imaged with the confocal microscope.

## Confocal imaging

All images were acquired on LSM 710 and LSM 880 Zeiss microscopes using the following laser wavelengths: argon 458/488/514, HeNe 543, and HeNe 633, along with the corresponding filter: MBS 488/543/633, MBS 458/543, MBS 488/543/633, using either a ×40 1.0 NA water immersion objective or a ×63 1.40 NA oil immersion objective. Confocal settings were adjusted to achieve maximum visualization of labeling at CE contact areas. Both lateral and 'en face' views of expanded CE contact areas were used for quantitative analysis. The lateral and axial resolutions of our system were estimated following the microscope specifications (*Wilhelm et al., 2010*). We estimated the lateral and axial resolutions of our microscope (*Wilhelm et al., 2010*) to range between 248.9 to 322 nm and 429.44 to 557.04 nm, respectively, depending on the excitation wavelength used (488, 543, 633). After correcting for the linear expansion factor (3.9×), the lateral and axial resolutions after expansion are expected to range between 63.81 to 82.77 nm and 110.11 to 142.83 nm, respectively.

## Quantification and statistical analysis

Confocal images were obtained using ZEN (black edition) software and analyzed with FIJI. The scale bars in the figures represent actual dimensions, and, therefore, ProExM images have not been adjusted for expansion factor. Contrast and brightness of fluorescence channels were individually adjusted and contrasted in Photoshop (Adobe) using blur and sharpen filters. Nonetheless, quantitative analysis of image fluorescence was carried out using raw data.

## Colocalization analysis

Colocalization analysis was performed in FIJI (https://imagej.net/imaging/colocalization-analysis) using the JACoP plugin (https://imagej.net/plugins/jacop, which uses pixel-wise methodology to determine matching pixels between channels). After selecting ROIs of the oval CE contact areas, colocalization for each image was quantified using the Manders' coefficient (*Bolte and Cordelières, 2006*; *Zinchuk et al., 2007*). This coefficient expresses the proportion of fluorescence in one channel that colocalizes with the proportion of fluorescence in the second channel, and vice versa (*Manders et al., 1993*). The analysis allowed for setting independent thresholds for each channel to account for different levels of fluorescent intensity and generate colocalization coefficients whose values range from 0 to 1.

## Center/periphery analysis

To quantify the differential distribution of GluR2 and Cx35.5 labeling (*Figure 2*), we defined two ROIs at 'en face' views of the CE contact: a central oval ROI representing ¾ of the area ('center') and an annular ROI representing the peripheral remaining ¼ ('periphery'). To estimate the surface area covered by GluR2 or Cx35.5 labeling, we determined the fluorescence intensity of each channel for both the central and peripheral ROIs and obtained the 'area integrated intensity' (sum of the pixel intensity over all of the pixels in the ROI). Background was then subtracted using the Correct Total Cell Fluorescence [CTCF = Integrated Density – (Selected area. Background mean fluorescence)]. For statistical comparison, CTCF values in the 'center' and 'periphery' ROIs were normalized to the highest values in each case, which, because of their differential distribution, in most of the cases were Cx35.5 in the center and GluR2 in the periphery. Statistical comparison was performed using Student's *t*-test.

## Labeling occupancy at CE contact area

The area of occupancy of the labeling for different synaptic proteins was performed in FIJI using the protocol developed by Jacqueline Ross (University of Auckland, New Zealand). After selecting an ROI

of the oval CE contact area, the procedure identifies fluorescent patches to then calculate the total labeled area for each florescence channel (for more details on this method, please see here).

For estimates of Cx35.5 puncta area (*Figure 5* and related text), each punctum was manually defined as an ROI, and the distribution of the estimated area values was plotted as frequency histograms (the number of bins was selected using Sturge's rule).

## Acknowledgements

We thank Martin Pinter, Anne Martin, Adam Miller, and members of the Pereda lab for critical feedback on the work and manuscript. We also thank Martin Pinter for his help on the statistical analysis of the expansion method.

## Additional information

### Funding

| Funder | Grant reference number | Author |
|---|---|---|
| National Institute on Deafness and Other Communication Disorders | R01DC011099 | Alberto E Pereda |
| National Institute of Neurological Disorders and Stroke | R21NS085772 | Alberto E Pereda |
| National Institute of Mental Health | RF1MH120016 | Alberto E Pereda |

The funders had no role in study design, data collection and interpretation, or the decision to submit the work for publication.

### Author contributions

Sandra P Cárdenas-García, Conceptualization, Data curation, Formal analysis, Methodology, Writing - original draft, Project administration, Writing - review and editing; Sundas Ijaz, Conceptualization, Formal analysis, Methodology, Writing - review and editing; Alberto E Pereda, Conceptualization, Supervision, Funding acquisition, Writing - original draft, Project administration, Writing - review and editing

### Author ORCIDs

Sandra P Cárdenas-García ⓘ http://orcid.org/0000-0002-7001-4446
Sundas Ijaz ⓘ https://orcid.org/0009-0005-8199-7598
Alberto E Pereda ⓘ https://orcid.org/0000-0002-8283-8768

### Ethics

This study was performed in strict accordance with the recommendations in the Guide for the Care and Use of Laboratory Animals of the National Institutes of Health. All of the animals were handled according to approved institutional animal care and use committee (IACUC) protocols (#00001029) of the Albert Einstein College of Medicine.

### Decision letter and Author response

Decision letter https://doi.org/10.7554/eLife.91931.sa1
Author response https://doi.org/10.7554/eLife.91931.sa2

## Additional files

### Supplementary files

- MDAR checklist
- Supplementary file 1. Key resources table.

## Data availability

All data generated and analyzed for Figures 2D-F; 3E-F; 4C-F; 5D-E; 6B,D, F; 7A-D; and Figure 1—figure supplement 1B is available as source data Excel files on G-Node (https://doi.org/10.12751/g-node.8ljsii).

The following dataset was generated:

| Author(s) | Year | Dataset title | Dataset URL | Database and Identifier |
|---|---|---|---|---|
| Cárdenas-García SP, Ijaz S, Pereda AE | 2024 | The components of an electrical synapse as revealed by expansion microscopy of a single synaptic contact | https://doi.org/10.12751/g-node.8ljsii | G-Node GIN, 10.12751/g-node.8ljsii |

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
