## [Editor Report]

This manuscript provides fundamental insights into how components of an electrical synapse are arranged at identified gap junctions using expansion microscopy. They provide convincing evidence for how these molecular components are placed within the junction. Such analysis is important for our understanding of synaptic organization and function.

---

## [Decision Letter]

**Decision letter after peer review:**

Thank you for submitting your article "The components of an electrical synapse as revealed by expansion microscopy of a single synaptic contact" for consideration by *eLife*. Your article has been reviewed by 3 peer reviewers, one of whom is a member of our Board of Reviewing Editors, and the evaluation has been overseen by Sacha Nelson as the Senior Editor.

Essential revisions (for the authors):

1) As detailed in the individual reviewer comments below, please elaborate on the methodological details.

2) Also include quantification of key results as mentioned by Reviewer #3.

*Reviewer #1 (Recommendations for the authors):*

In this manuscript, authors use expansion microscopy to map the localization of components of electrical and chemical synapses in a well-studied mixed synapse. Mauthner neurons, the command-like neurons of the teleost escape circuit, receive mixed synapses from auditory afferents. In the present study, authors find that at these synapses, the chemical synaptic components are relegated to the rim while the bulk of the contact is occupied by gap junctions and adherens junctions. They also find that the electrical synapse is made up of numerous gap junctional contacts, variable in number and size within and across CEs. These findings are significant because such detailed maps of electrical synapses and their relationships to chemical synapses have not been made. In addition, the study underlines the importance of adherens junctions for the maintenance of gap junctions. Having said that there are a few comments that I'd like the authors to address:

1. Authors use the Cx35 and GluR2 staining to ascribe the borders of the CE. Why couldn't they have used a cytoplasmic fill of the CE (gal4 y256 or retrograde label) to more accurately demarcate the CE boundary? This would also help them estimate the expansion factor more accurately before and after expansion in the same sample.

2. In both the non-expanded and the expanded images of Figure 1 as well as in other figures, the Cx35 staining is intense on one side of the CE (see Figure 1C and E for example). Yet, the quantification shows that the central staining for Cx35 is stronger than the periphery. It is also not clear how the intensity normalization was done for Figure 2E and F. I could not understand Figure 2F. The legend for this panel is also very vague. In general, the analysis methods need to be described in greater detail.

3. What does n represent? Number of CE's sampled or number of fish? Better to include both.

4. One question is whether the organization of GJs and AJs is universal. Have the authors examined other GJ plaques on the M cell (such as those seen in Figure 1B) to see if they are surrounded by AJs? Can also look at these for the colocalization of other markers such as Cx34 and ZO.

5. In Figure 7A authors use ratios of fluorescence intensity to argue that Cx and AJs are roughly equal. However, there are several caveats since antibody staining is used (relative affinities of primary and secondary antibodies, the brightness of fluorophores etc.,) It is not clear if these have been accounted for.

*Reviewer #2 (Recommendations for the authors):*

In this manuscript, Cárdenas-García et al. describe the structure and organization of gap junction electrical synapses at specialized synapses, large myelinated club endings (CEs) in larval zebrafish. They find, using super-resolved expansion microscopy, structures resembling gap junction plaques at the CEs. They find these plaques are arranged independently from glutamate receptors and seem to be connected to other cell-cell junctions, specifically adherens junctions. These plaques exist in a range of sizes, which they speculate may play a role in the coordination of electrical transmission across these synapses.

Methods:

The use of expansion microscopy for subcellular mapping and probing the organization of gap junctions at these synapses is appropriate- however, a lot of controls and methodological information are missing making it impossible to gauge how reliably the expansion worked.

Important missing information:

While not clearly labeled, presumably Figure 1B-D represent "pre/non-expanded" state and Figure 1E represents a post-expansion state. For expansion microscopy, it's standard practice the estimate the expansion factor, by measuring the gel before and after/measuring the distance between two known structures/or imaging the same cell(s) before and after expansion. Without this information, it's difficult to assess if they've achieved sufficient resolution to make the structural organization claims they've outlined. Additionally in the methods, they outline stock solutions typically used in ProExpansion, rather than the actual concentrations in the monomer solution used to pour the gels. If the experiments were performed as stated in the methods, with no expansion in water but just in an overnight proteinase K digestion buffer- this would achieve only around 1.5x expansion- far below the amount needed to sufficiently map subsynaptic distributions of these proteins.

Analysis:

The use of line scans for mapping the distributions of different proteins at the synapse is the gold standard. However, a lot of co-localization analysis is included- including on structures which clearly show zero overlap in their representative images. The details of this analysis are sparse, only referencing a fiji plugin, with no perimeters used in this analysis provided.

While potentially a useful resource for those working on electrical synapses, with these missing controls and information for the expansion process itself, and questionable analysis; the robustness of the results and conclusions are difficult to properly assess.

Overall figure suggestions:

Changing colours for LUTs between figures is distracting, and since at times they use Green/Magenta which is more accessible for anyone with visual impairments or colour blindness, should be used throughout the manuscript.

Figure 1:

– Cx35 and Cx35.5 are used interchangeably, and presumably are the same?

– Is panel D coming from this same example in B? If not, how it's arranged makes it seem like it would.

– Why does the distribution of Cx35.5 appear different in B and E? Is B a preexpansion image? All the legend says is "confocal" image for B, so presumably the authors mean "non-expanded".

– What is the expansion factor achieved?

– The expansion protocol listed in the methods isn't complete- it's just listing the stock concentrations of that would be used for the monomer solution. It's referenced that there are additional modifications- but these aren't listed or stated. What are they?

–With ProExpansion, you don't get more than 1.5x expansion in the proteinase K digestion buffer.. was this really all that was performed and no further expansion in water was carried out?

Figure 2:

– Co-localization seems inappropriate for this sort of analysis, especially if the data is representative of the images shown in the figure. And the arbitrary point analysis in E and F also seems to be an odd choice. Line scan analysis would provide far more relevant information regarding the distribution of Cx35.5 and GluR2- which would help at the end build a molecular map of the distributions of these proteins at the electrical synapse.

"En face" view is mentioned, what exactly is meant by this?

Figure 3:

– The co-localization analysis is a lackluster way of assessing the distribution of these proteins, especially with the much more relevant and appropriate line scan analysis being performed in Figure 4.

Figure 4:

– Showing a Cx34.1 – ZO1 combination and analysis would strengthen the claims here.

– If the authors could quantify and report the expansion factor they achieve, it would better allow the reader to understand what the distances in F actually would be in the pre-expanded "native" state.

Figure 5:

Was this analysis really only performed on three cells? Was it from a single experiment?

Figure 6:

– The line scan analysis is more meaningful than the co-localization analysis here, it would help if controls including N-Cadherin vs B-Catenin were shown (since they would be in the same complex).

–"En face" view is mentioned again, what exactly is meant by this?

Figure 7:

Pie charts which aren't labelled for their values, seem an odd choice. How this analysis is performed, and what is actually being quantified is really unclear. It would help for more methodological details and explanations to be provided.

*Reviewer #3 (Recommendations for the authors):*

Summary: In this manuscript, Cárdenas-García at el. investigated the nanoscale structure and molecular composition of electrical synapses in 5 days post fertilization zebrafish larvae. The study revealed the relative distribution of various connexin proteins at electrical synapses at Club endings and the potential contribution of additional proteins, such as N-cadherin and ß-catenin, to the function of electrical synapses.

Strengths: The authors employed powerful and validated imaging techniques, enabling multi-color super-resolution imaging of thick tissue samples. Using this technique, the authors revealed the relative localization and distribution of several electric synapse components.

Weakness: I think the authors did not take full advantage of the imaging technique used. ExM allows for volumetric super-resolution imaging using confocal microscopy, however, the authors did not generate any 3D images of the electrical synapses, which may facilitate 3D structural analysis in more detail. Also, it is unclear what effective spatial resolution was achieved for the provided images; this knowledge is essential when interpreting the results. In addition, the manuscript suffers from many vague and misleading statements regarding results descriptions, for example, "Our data suggest that synaptic communication at electrical synapses results from not one, but the coordinated action of multiple GJs of variable size" – how variable? Why is it not quantified? There are many other similar statements missing quantitative information that can be easily obtained from simple image analysis.

While the manuscript structure is well organized, some further editing is needed to improve the quality of writing and strengthen the statements.

1) The manuscript is missing a lot of critical technical details (e.g., effective imaging resolution, linear expansion factor of expanded samples for each image in figure legends, scale bars were corrected to expansion factor or not) and proper descriptive statistics that would be important for the reproducibility of the results. The authors never provide a description for n. Is it the number of fish imaged? Please report full descriptive statistics including the number of Club Ending imaged from how many fish.

2) Probably the most weak aspect of this manuscript is that many statements are vague due to the lack of quantitative information that authors can easily obtain by simple image analysis. For example, "Thus, expansion of these single synapses resulted in a more than 10-fold increase of the synaptic contact area, allowing for a more detailed visualization of the relative distribution of its synaptic components" – a more detailed visualization can be easily specified by providing effective lateral and axial resolution (simply divide the resolution of achieved with respective imaging settings by expansion factor). There are many similar examples throughout the manuscript. There are just so many vague statements like this one that I may not be able to list all of them here, but here are some more example:

Multiple gap junctions of variable size were identified by the presence of their molecular components. – How variable? Can it be expressed quantitatively? From X to Y um?

Because of their unusually large size and experimental accessibility, CEs represent a valued model for the correlation of synaptic structure and function. – How large is large?

Which are revealed as large fluorescent oval areas at the distal portion of the lateral dendrite of the M-cell (Figure 1 B-D). – can it be written as ~1-um fluorescent oval instead?? In Figure 1E I see 5-um structures. Are they bigger than large?

Which are more difficult to obtain because of their larger size. – How much larger?

high degree of colocalization (Figure 3B,D) – please provide a numeric expression of the lower upper range in the text in addition to data in Figure 3B,D.

Please go over the rest of the manuscript and fix the qualitative descriptions where possible.

3) The first sentence of the introduction is so vague and misleading, with inadequate citation that it should be either removed or completely rewritten. The way it is written suggests that chemical synapses are bidirectional, synapses can be either chemical or electrical but not both, synapses occur only between neurons. The citation is misleading, too.

4) The introduction is so general without proper specification of species the authors are referring to, it sounds like electrical synapses are the same in all organisms. Is it actually the case?

5) Is it possible to provide one or two Supplementary Video visualizing 3D structure of electric synapses using ExM images? Or all images provided in the manuscript were single z-plane?

6) Images in Figures were generated as max projection, or are they single plane, this information should be provided for every single image shown in the manuscript indicating z-step size.

7) Although a distance between Cx35.5 and Cx34.1 labeling peaks was observed in some examples, it was still too small to be detected with this method due to the spatial amplification produced by the fluorophores. – How small is too small? What is spatial amplification?

8) The observed distance between the peaks of fluorescence was not due to the differences in the wavelength of the fluorophores, and it remained when secondary antibodies were inverted (Figure 4D-E). – Do you mean achromatic aberration? If so, why should it be different when switching secondary antibodies? How achromatic aberration was corrected for the images?

---

## [Author Response]

Essential revisions (for the authors):1) As detailed in the individual reviewer comments below, please elaborate on the methodological details.2) Also include quantification of key results as mentioned by Reviewer #3.

We have followed the Reviewing Editor’s requests and fully address the reviewers’ comments. See below.

Reviewer #1 (Recommendations for the authors):In this manuscript, authors use expansion microscopy to map the localization of components of electrical and chemical synapses in a well-studied mixed synapse. Mauthner neurons, the command-like neurons of the teleost escape circuit, receive mixed synapses from auditory afferents. In the present study, authors find that at these synapses, the chemical synaptic components are relegated to the rim while the bulk of the contact is occupied by gap junctions and adherens junctions. They also find that the electrical synapse is made up of numerous gap junctional contacts, variable in number and size within and across CEs. These findings are significant because such detailed maps of electrical synapses and their relationships to chemical synapses have not been made. In addition, the study underlines the importance of adherens junctions for the maintenance of gap junctions.

We thank the reviewer for his/her comments on the significance of our findings.

Having said that there are a few comments that I'd like the authors to address:1. Authors use the Cx35 and GluR2 staining to ascribe the borders of the CE. Why couldn't they have used a cytoplasmic fill of the CE (gal4 y256 or retrograde label) to more accurately demarcate the CE boundary? This would also help them estimate the expansion factor more accurately before and after expansion in the same sample.

The contact areas of the Club endings (CEs) can be reliably defined by the labeling of gap junction (GJ) proteins, as these structures are homogenously distributed throughout the contact area. We previously showed this feature by simultaneously labeling presynaptic afferents in goldfish (Flores et al., 2010; Pereda et al., 2003), and more recently, in zebrafish (Yao et al., 2014), a finding which is consistent with freeze-fracture electron microscopy reconstruction of these terminals (Tuttle et al., 1986). Figure 1 illustrates how Cx35 labeling nicely matches the contact area of the Club endings of labeled auditory afferents (from Flores et al., 2010). Thus, we feel confident that the expansion factor can be estimated by measuring the dimensions of these highly stereotyped oval-shaped labeled areas. To better illustrate this feature, we have now modified the cartoon of Figure 1A, which we hope will help the reader with the interpretation of the labeled areas. The reviewer’s suggestion is however well-taken, and the use of the gal4-y256 line will become extremely useful when analyzing fish with mutations in genes that code for GJ-associated proteins, at which the distribution of connexins might be altered. We thank the reviewer for this suggestion.

2. In both the non-expanded and the expanded images of Figure 1 as well as in other figures, the Cx35 staining is intense on one side of the CE (see Figure 1C and E for example). Yet, the quantification shows that the central staining for Cx35 is stronger than the periphery. It is also not clear how the intensity normalization was done for Figure 2E and F. I could not understand Figure 2F. The legend for this panel is also very vague. In general, the analysis methods need to be described in greater detail.

We thank the reviewer for noticing this inconsistency. While we use ‘Fluorescence intensity’ in the labels of the graphs we are actually measuring the presence of fluoresce over area, regardless of its intensity. Unfortunately, the wrong and therefore misleading labels created a confusion in the interpretation of our analysis. We apologize for this mistake. We describe now, in greater detail, our method of analysis (see Methods section; page 20). We have also modified the labels in Figures 4,6, and 7, and associated text to make clear that we are measuring fluoresce over area, regardless of its intensity.

3. What does n represent? Number of CE's sampled or number of fish? Better to include both.

We thank the reviewer for noticing this. We realize now that we didn’t clearly define the meaning of ‘n’ in each data set, which depends on the type of experiment. In most cases an ‘n’ represents an ‘En face’ view of a CE contact, on which analysis of fluorescence and co-localization over the contact area of the terminal can be performed. In contrast to adult animals, ‘En face’ views are harder to obtain in larval zebrafish because the diameter of the CE contact is similar to the diameter of the lateral dendrite of the Mauthner cell (Yao et al., 2014), which, together, with the presence of a smaller number of these afferents, makes ‘En face’ views less likely to be found and measured. In other cases ‘n’ represents labeled puncta. As requested, we now provide both the number of ‘En face’ CE contacts or puncta, and the number fish for each type of experiment. Please see highlighted text in relevant data sections throughout the paper.

4. One question is whether the organization of GJs and AJs is universal. Have the authors examined other GJ plaques on the M cell (such as those seen in Figure 1B) to see if they are surrounded by AJs? Can also look at these for the colocalization of other markers such as Cx34 and ZO.

We thank the reviewer for bringing up this important point. Ongoing work in the lab focuses on describing the association between these two structures (GJs and AJs) to define the boundaries of electrical synapses in synaptic and cell-cell contacts elsewhere, with the goal of exposing their variability (i.e.: the presence of different numbers of GJs) as a potential source of synaptic diversity. This analysis will be the focus of a separate study, documenting the prediction of this association. However, to address the reviewer's concern, and although it will not be included in the present paper, we show here labeling for Cx35.5 and N-Cadherin at small unidentified synaptic contacts nearby the CEs terminal field on the lateral dendrite of the M-cell. Just as CEs, most excitatory inputs to the Mauthner cell are also mixed (electrical and chemical) and thus, smaller terminals can also be exposed by the presence of Cx35.5 labeling (Pereda et al., 2003; Rash et al., 2015; Yao et al., 2014). As observed in Figure 2, although smaller, these terminals exhibit the same pattern of association between the labelings. Also supporting the ubiquity of the relationship between GJs and AJs, this association was also observed in mammals (see Discussion section; page 17). Nagy and Lynn, 2018 show associated, but mutually exclusive labeling for Cx36 and N-cadherin at somato-somatic contacts between cells of the mesencephalic nucleus of the trigeminus (MesV) in mice, in a pattern that resembles the association we report at CEs in our study.

5. In Figure 7A authors use ratios of fluorescence intensity to argue that Cx and AJs are roughly equal. However, there are several caveats since antibody staining is used (relative affinities of primary and secondary antibodies, the brightness of fluorophores etc.,) It is not clear if these have been accounted for.

As mentioned above, we clarify now that our data represents the measurement of fluorescence over area, independent of its intensity. We agree with the caveats listed by the reviewer if we were reporting intensity, not fluorescence over area, regardless of its intensity. We have clarified this in Figure 7A and elsewhere.

Reviewer #2 (Recommendations for the authors):In this manuscript, Cárdenas-García et al. describe the structure and organization of gap junction electrical synapses at specialized synapses, large myelinated club endings (CEs) in larval zebrafish. They find, using super-resolved expansion microscopy, structures resembling gap junction plaques at the CEs. They find these plaques are arranged independently from glutamate receptors and seem to be connected to other cell-cell junctions, specifically adherens junctions. These plaques exist in a range of sizes, which they speculate may play a role in the coordination of electrical transmission across these synapses.

We thank the reviewer for the summary of our findings.

Methods:The use of expansion microscopy for subcellular mapping and probing the organization of gap junctions at these synapses is appropriate- however, a lot of controls and methodological information are missing making it impossible to gauge how reliably the expansion worked.

We provide, in this revised version, the requested additional information for a better assessment of the findings of the paper.

Important missing information:While not clearly labeled, presumably Figure 1B-D represent "pre/non-expanded" state and Figure 1E represents a post-expansion state. For expansion microscopy, it's standard practice the estimate the expansion factor, by measuring the gel before and after/measuring the distance between two known structures/or imaging the same cell(s) before and after expansion. Without this information, it's difficult to assess if they've achieved sufficient resolution to make the structural organization claims they've outlined.

We agree with the reviewer that, at the early stages of the development of the expansion procedure, it was a standard practice to measure the expansion of the gel in each experiment. This macroscopic estimate can still be useful in some cases where the organization of the targeted microscopic structures is unknown. In other words, while coarse, a macroscopic estimate of the size of the gel could provide some confidence on the interpretation of unknown, expanded, microscopic structures. However, because of its limited accuracy, this practice became unnecessary, especially when the organization of the targeted structure is known and sufficiently recognizable prior to expansion. See, for example, the study by Zhu et al. (Zhu et al., 2021), in which they state: "The macroscopic measurement simply measures and compares the sizes of the hydrogel before and after expansion [7], which, however, may not truly represent the microscopic expansion of the tissue or cellular structures of interest [8, 9]". See also, along these lines, the studies of Kubalova et al. (Kubalová et al., 2020), and Damstra et al. (Damstra et al., 2022), the latter published in this journal.

Easily identifiable because of their unusual large size (2 µm in larval ZF and 10 µm in adult goldfish), the auditory terminals known as CEs have historically provided the opportunity to explore their synaptic organization with a variety of technical approaches. Electron microscopy (EM) of these terminals provided the first evidence for the role of GJs as the basis for electrical transmission (Robertson et al., 1963), and freeze-fracture EM revealed their overall organization in goldfish (Tuttle et al., 1986), as well as the molecular composition of their GJs in goldfish and zebrafish (Pereda et al., 2003; Rash et al., 2013; Yao et al., 2014). The known structural features of CEs make them ideal for anatomical measurements using immunolabeling. That is, labeling for GJ proteins (connexins and ZO1) at these terminals was shown to faithfully reproduce the contact surface area of the terminal (see Figure 1 of this document). Moreover, these highly stereotyped oval labeled areas were shown to be highly consistent amongst different terminals and easily identifiable in adult goldfish (Flores et al., 2010, 2008; Pereda et al., 2003; Rash et al., 2013) as well as in adult and larval zebrafish of various ages by us (Yao et al. 2015) and colleagues (Jabeen and Thirumalai, 2013; Miller et al., 2017; Wolman et al., 2015).

Thus, we feel confident that the diameter and shape of the contact area labeled for GJ proteins constitutes a truthful representation of the pre-expanded terminal size, and therefore, a finer and more accurate estimate of its expanded dimensions than the coarse macroscopic estimate provided by gel measurement. Moreover, the highly consistent dimensions of these terminals also renders it unnecessary to estimate their size before expansion, and the expansion of the gel in each experiment. As shown in Supp Figure 1 and related text, measurements of the diameters of these two-dimensional oval contacts allowed us to expose a high-degree of isometry in our expansions, we argue, with much better accuracy than that eventually predicted by measurement of gel expansion.

Additionally in the methods, they outline stock solutions typically used in ProExpansion, rather than the actual concentrations in the monomer solution used to pour the gels. If the experiments were performed as stated in the methods, with no expansion in water but just in an overnight proteinase K digestion buffer- this would achieve only around 1.5x expansion- far below the amount needed to sufficiently map subsynaptic distributions of these proteins.

The reviewer is correct. We realized that, while obvious, we omitted to include the addition of water in the description of the technique. We apologize for the omission and thank the reviewer for noticing it. We have now included the addition of water in the methods. As suggested by the reviewer, we now list the actual concentrations in the monomer solution used to pour in the gels. Finally, we have expanded the general description of the Methods section, which will also be available in more detail in BioProtocols (see the submitted protocol at the end of this response).

Analysis:The use of line scans for mapping the distributions of different proteins at the synapse is the gold standard. However, a lot of co-localization analysis is included- including on structures which clearly show zero overlap in their representative images. The details of this analysis are sparse, only referencing a fiji plugin, with no perimeters used in this analysis provided.

We used the line scan when we judged it appropriate, see for example, Figures 4 and 6, in order to show the presence or absence of co-localization of two neighboring labeled structures. However, in some other experiments of this study, the goal was to measure the amount of fluoresce over area to generate a map of the spatial distribution and relative amounts of different synaptic components within the contact. The line scan will not be useful for this purpose, as it only provides information about the presence or absence of fluorescence along the straight line. With respect to the use of reporting lack of co-localization, we believe that there is potentially as much information in the presence, as there is in the absence of co-localization, as labeled proteins can be part of the same or different sub-cellular structures. In other words, its use depends on the experimental question. Finally, in response to the reviewer’s concern, we now provide detailed information regarding our data analysis and the parameters used for the Fiji plugin. Colocalization analysis was performed in Fiji using the JACoP plugin (pixel-wise methodology to determine matching pixels between channels). The JACoP plugin has also been utilized and validated by another group in immunostained zebrafish embryos, followed by Label Retention Expansion Microscopy (LR-ExM) to analyze the colocalization of proteins (Zhao et al., 2022, 2021). See page 20 of the revised manuscript.

While potentially a useful resource for those working on electrical synapses, with these missing controls and information for the expansion process itself, and questionable analysis; the robustness of the results and conclusions are difficult to properly assess.

Please see our response above. We hope we now provide the information required for a better assessment of the paper.

Overall figure suggestions:Changing colours for LUTs between figures is distracting, and since at times they use Green/Magenta which is more accessible for anyone with visual impairments or colour blindness, should be used throughout the manuscript.

We appreciate the reviewer’s comment about the choice of colors in the figures. We initially tested the use of green and magenta, but we found that it wasn’t as helpful in showing the presence of colocalization when using double-labeling. More importantly, magenta was not as effective as red in capturing the tridimensional features of the labeled areas. Thus, while we truly appreciate the needs of those with color impairment, it is because of the above reasons that we decided to: (1) use red and green to show the absence or presence of co-localization of various proteins at the electrical synapse regardless of the label used, (2) use magenta only for glutamate receptor labeling with the aim of clearly differentiating the chemically transmitting area from the components of the electrical synapse, which were labeled, again, only with either red or green. We hope the reviewer understands and agrees with the rationale of the selected color scheme.

Figure 1:– Cx35 and Cx35.5 are used interchangeably, and presumably are the same?

We thank the reviewer for pointing this out. We apologize for the confusion. In the previous version of the manuscript, we used ‘Cx35’ to indicate the use of Cx35/36 antibody, which recognizes both Cx35.1 and Cx35.5, and we used ‘Cx35.5’ to indicate the use of the Cx35.5-specific antibody. We now make this distinction clear, indicating which antibody was used. Also, since only Cx35.5 was reported to be present at these terminals, to avoid confusion, we now use ‘Cx35.5’ throughout the paper, indicating, in each case, if it was labeled with either the Cx35/36 antibody or the Cx35.5-specific antibody.

– Is panel D coming from this same example in B? If not, how it's arranged makes it seem like it would.

Panel D does not come from the example in B, it comes from a different experiment, in which we instead used the ZO1 antibody. We chose to display panels C (Cx labeling) and D (ZO1 labeling) side by side to better appreciate the consistency and similarity of labeling for different GJ proteins, either channels or scaffolds, in non-expanded terminals.

– Why does the distribution of Cx35.5 appear different in B and E? Is B a preexpansion image? All the legend says is "confocal" image for B, so presumably the authors mean "non-expanded".

We thank the reviewer for pointing out this ambiguity in the labels. We have now added the labels ‘non-expanded’ in panel B to indicate that, in panels B, C, and D, labeling was not followed by expansion, and ‘ProExM’ in panel E to indicate that the tissue is expanded. Also, while panels B and C are from the same experiment. we realized that panel C appears, by mistake, in a different orientation. We have now horizontally flipped panel C for consistency.

– What is the expansion factor achieved?

We added the expansion factor to the figure and text, which we estimated to be 3.9x in diameter and 13.4x in area (see legend of Figure 1 and page 6).

– The expansion protocol listed in the methods isn't complete- it's just listing the stock concentrations of that would be used for the monomer solution. It's referenced that there are additional modifications- but these aren't listed or stated. What are they?–With ProExpansion, you don't get more than 1.5x expansion in the proteinase K digestion buffer.. was this really all that was performed and no further expansion in water was carried out?

See response above.

Figure 2:– Co-localization seems inappropriate for this sort of analysis, especially if the data is representative of the images shown in the figure. And the arbitrary point analysis in E and F also seems to be an odd choice. Line scan analysis would provide far more relevant information regarding the distribution of Cx35.5 and GluR2- which would help at the end build a molecular map of the distributions of these proteins at the electrical synapse."En face" view is mentioned, what exactly is meant by this?

We have now clarified our method of analysis (see Methods section; page 20). In this experiment our goal was to measure the amount of fluoresce over area to create a map of the spatial distribution and relative amounts of the synaptic components of electrically vs. chemically mediated areas within the contact. The line scan in this case will not be useful, as it will only provide information about the presence of fluorescence along a straight line. See response above.

‘En face’ means ‘facing forward’, such as in a human portrait. The term is classically used in anatomy to describe images of structures facing forward. In our case, we use it to describe, in contrast to side views, images of CEs facing forward so their complete contact area can be observed and analyzed.

Figure 3:– The co-localization analysis is a lackluster way of assessing the distribution of these proteins, especially with the much more relevant and appropriate line scan analysis being performed in Figure 4.

See above our response regarding the use of line scan and co-localization analysis.

Figure 4:– Showing a Cx34.1 – ZO1 combination and analysis would strengthen the claims here.

We agree with the reviewer. Unfortunately, the antibodies against Cx34.1 and ZO1 are both monoclonal and incompatible for use together, which prevented us from doing this experiment. However, we plan to do this experiment in the future, as soon as compatible antibodies and/or alternative approaches become available.

– If the authors could quantify and report the expansion factor they achieve, it would better allow the reader to understand what the distances in F actually would be in the pre-expanded "native" state.

We more clearly report the expansion factor in page 6. The linear expansion factor was 3.9x, which led to a 13.4x increase in the area of the contact.

Figure 5:Was this analysis really only performed on three cells? Was it from a single experiment?

No, it comes from several experiments, we apologize for the confusion. This labor intensive analysis summarizes the data obtained from 3 dendrites, each belonging to a different fish (bar graphs labeled as ‘Dendrite Cell 1’, ‘Dendrite Cell 2’, and ‘Dendrite Cell 3’ in panel D). Each histogram in panel D illustrates, overlapped in different shades of blue, the values obtained from the analysis of three ‘En face’ views of CE terminals at a single dendrite, which, as discussed above, are hard to find, especially in the same dendrite after expansion. Finally, the frequency histogram in E summarizes the values resulting from the analysis of these nine ‘En face’ CE views. We have clarified this in the main text (see page 9) and Figure 5 legend.

Figure 6:– The line scan analysis is more meaningful than the co-localization analysis here, it would help if controls including N-Cadherin vs B-Catenin were shown (since they would be in the same complex).

As we discuss above, we believe that the line scan and the co-localization analyses provide complementary information, and therefore we would like to keep both.

–"En face" view is mentioned again, what exactly is meant by this?

See response above.

Figure 7:Pie charts which aren't labelled for their values, seem an odd choice. How this analysis is performed, and what is actually being quantified is really unclear. It would help for more methodological details and explanations to be provided.

We thank the reviewer for the suggestion. Now we include the values in the pie-charts and also include non-labeled areas, which, when left out, led to a misrepresentation of the labeled proportions. We now clarify that what is quantified is the relative amounts of fluorescence over area. For this purpose, we have now expanded and clarified the method of analysis (see page 20 of the revised manuscript).

Reviewer #3 (Recommendations for the authors):Summary: In this manuscript, Cárdenas-García at el. investigated the nanoscale structure and molecular composition of electrical synapses in 5 days post fertilization zebrafish larvae. The study revealed the relative distribution of various connexin proteins at electrical synapses at Club endings and the potential contribution of additional proteins, such as N-cadherin and ß-catenin, to the function of electrical synapses.Strengths: The authors employed powerful and validated imaging techniques, enabling multi-color super-resolution imaging of thick tissue samples. Using this technique, the authors revealed the relative localization and distribution of several electric synapse components.

We thank the reviewer for the summary of our findings and their significance.

Weakness: I think the authors did not take full advantage of the imaging technique used. ExM allows for volumetric super-resolution imaging using confocal microscopy, however, the authors did not generate any 3D images of the electrical synapses, which may facilitate 3D structural analysis in more detail. Also, it is unclear what effective spatial resolution was achieved for the provided images; this knowledge is essential when interpreting the results. In addition, the manuscript suffers from many vague and misleading statements regarding results descriptions, for example, "Our data suggest that synaptic communication at electrical synapses results from not one, but the coordinated action of multiple GJs of variable size" – how variable? Why is it not quantified? There are many other similar statements missing quantitative information that can be easily obtained from simple image analysis.

We appreciate the reviewer’s concerns and his/her very useful feed-back on the paper. We address each of these concerns below. In brief, we realized that we were not sufficiently explicit about the scale bars representing the expanded dimensions, which in turn led to confusion regarding the effective spatial resolution of our data. Also, we have made efforts to clarify various statements along the text. Finally, although we appreciate that expansion microscopy can be used to generate 3D images of the electrical synapses, this analysis was not necessary for the goals of this paper which was centered on generating a two-dimensional map of the contact’s surface area. We are planning to use 3D images of the electrical synapses in future studies, which will be centered on the association of pre- vs postsynaptic proteins and/or the proximity of associated structures, such as trafficking vesicles and mitochondria to electrical synapses, which would benefit from 3D analysis.

While the manuscript structure is well organized, some further editing is needed to improve the quality of writing and strengthen the statements.1) The manuscript is missing a lot of critical technical details (e.g., effective imaging resolution, linear expansion factor of expanded samples for each image in figure legends, scale bars were corrected to expansion factor or not) and proper descriptive statistics that would be important for the reproducibility of the results. The authors never provide a description for n. Is it the number of fish imaged? Please report full descriptive statistics including the number of Club Ending imaged from how many fish.

We realize that we were not sufficiently explicit about that the scale bars representing the expanded dimensions, not normalized to the expansion factor, thus creating confusion regarding the effective spatial resolution of our images. We have clarified this throughout the main text, Methods section, and figure legends. In addition, we now provide a better description of the statistical analysis and the meaning of ‘n’ for each experiment. We have, in addition, fully re-analyzed our data, leading, in some cases, to small changes in previously reported values. See highlighted text in the relevant data sections throughout the paper.

2) Probably the most weak aspect of this manuscript is that many statements are vague due to the lack of quantitative information that authors can easily obtain by simple image analysis. For example, "Thus, expansion of these single synapses resulted in a more than 10-fold increase of the synaptic contact area, allowing for a more detailed visualization of the relative distribution of its synaptic components" – a more detailed visualization can be easily specified by providing effective lateral and axial resolution (simply divide the resolution of achieved with respective imaging settings by expansion factor).

We have revised our statements and include quantitative information when appropriate. As suggested, we now provide estimates of lateral and axial resolution divided by the expansion factor (see Methods section; page 20).

There are many similar examples throughout the manuscript. There are just so many vague statements like this one that I may not be able to list all of them here, but here are some more example:Multiple gap junctions of variable size were identified by the presence of their molecular components. – How variable? Can it be expressed quantitatively? From X to Y um?

Again, we have made efforts to clarify our statements. However, the sentence mentioned by the reviewer appears in the Abstract and the Discussion where, in general, results are summarized. A full and quantitative description of the variability in size of GJs can be found in Figure 5 and associated text, including correction for expansion, which allowed an estimate of the average number of GJ channels at these terminals. In other words, our statement is supported by a thorough quantitative analysis, which should be obvious to the reader of the entire paper.

Because of their unusually large size and experimental accessibility, CEs represent a valued model for the correlation of synaptic structure and function. – How large is large?which are revealed as large fluorescent oval areas at the distal portion of the lateral dendrite of the M-cell (Figure 1 B-D). – can it be written as ~1-um fluorescent oval instead?? In Figure 1E I see 5-um structures. Are they bigger than large?which are more difficult to obtain because of their larger size. – How much larger?high degree of colocalization (Figure 3B,D) – please provide a numeric expression of the lower upper range in the text in addition to data in Figure 3B,D.Please go over the rest of the manuscript and fix the qualitative descriptions where possible.

See response above. We have corrected qualitative descriptions.

3) The first sentence of the introduction is so vague and misleading, with inadequate citation that it should be either removed or completely rewritten. The way it is written suggests that chemical synapses are bidirectional, synapses can be either chemical or electrical but not both, synapses occur only between neurons. The citation is misleading, too.

We agree with the reviewer that the reference is wrong, a mistake probably made while selecting the reference from our reference manager program (Mendeley). We thank the reviewer for noticing it and apologize for this error. We have now replaced the reference. With respect to the sentence itself: the sentence aims to be a conceptual introduction to the mechanisms underlying the two main modalities of synaptic communication. We read the sentence multiple times and there is no reference to the directionality of transmission. Perhaps the reviewer was thinking of a different sentence. As requested by the reviewer, we have now reworded the sentence.

4) The introduction is so general without proper specification of species the authors are referring to, it sounds like electrical synapses are the same in all organisms. Is it actually the case?

Now we clarify these issues in the Introduction section.

5) Is it possible to provide one or two Supplementary Video visualizing 3D structure of electric synapses using ExM images? Or all images provided in the manuscript were single z-plane?

We thank the reviewer for the suggestion. We provide two supplementary videos of the same contact area of the CE synapse with either single (see Supplemental movie 1) or multiple labeling (see Supplemental movie 1).

6) Images in Figures were generated as max projection, or are they single plane, this information should be provided for every single image shown in the manuscript indicating z-step size.

Now we provide this information. See figure legends and relevant sections of the revised manuscript.

7) Although a distance between Cx35.5 and Cx34.1 labeling peaks was observed in some examples, it was still too small to be detected with this method due to the spatial amplification produced by the fluorophores. – How small is too small? What is spatial amplification?

The distance between the postsynaptic channel and the postsynaptic scaffold is shorter than the distance between the presynaptic channel and postsynaptic scaffold. Now we make this point more clear by substituting ‘small’ for ‘shorter’. With regards to ‘spatial amplification’, we are referring to the fact that secondary antibodies, because of their size and emitted fluorescence, spatially amplify the sequence they target on a specific protein. Now we clarify this point in the revised version of the paper.

8) The observed distance between the peaks of fluorescence was not due to the differences in the wavelength of the fluorophores, and it remained when secondary antibodies were inverted (Figure 4D-E). – Do you mean achromatic aberration? If so, why should it be different when switching secondary antibodies? How achromatic aberration was corrected for the images?

Yes, we refer to ‘chromatic aberration’. In optics, chromatic aberration is a failure of a lens to focus all colors to the same point, potentially creating, in our case, a confound in the distance between the labelings of two fluorophores. The chromatic aberration is corrected by using ‘achromatic’ objectives, as is the case in our confocal microscope. However, to be rigorous, we swapped the fluorophores to demonstrate that the detected distance is independent of them, just in case there was some uncorrected chromatic aberration in our system. We thank the reviewer for asking us to be more explicit about this manipulation. We have modified the text accordingly (see page 8).

References:

Damstra HGJ, Mohar B, Eddison M, Akhmanova A, Kapitein LC, Tillberg PW (2022) Visualizing cellular and tissue ultrastructure using Ten-fold Robust Expansion Microscopy (TREx). *ELife* 11.

Flores CE, Cachope R, Nannapaneni S, Ene S, Nairn AC, Pereda AE (2010) Variability of distribution of Ca(2+)/calmodulin-dependent kinase II at mixed synapses on the mauthner cell: colocalization and association with connexin 35. J Neurosci 30:9488–99.

Flores CE, Li X, Bennett MVL, Nagy JI, Pereda AE (2008) Interaction between connexin35 and zonula occludens^-1^ and its potential role in the regulation of electrical synapses. Proc Natl Acad Sci U S A 105:12545–50.

Jabeen S, Thirumalai V (2013) Distribution of the gap junction protein connexin 35 in the central nervous system of developing zebrafish larvae. Front Neural Circuits 7.

Kubalová I, Schmidt Černohorská M, Huranová M, Weisshart K, Houben A, Schubert V (2020) Prospects and limitations of expansion microscopy in chromatin ultrastructure determination. Chromosom Res 28:355.

Miller AC, Whitebirch AC, Shah AN, Marsden KC, Granato M, O’Brien J, Moens CB (2017) A genetic basis for molecular asymmetry at vertebrate electrical synapses. *ELife* 6.

Nagy JI, Lynn BD (2018) Structural and Intermolecular Associations Between Connexin36 and Protein Components of the Adherens Junction–Neuronal Gap Junction Complex. Neuroscience 384:241– 261.

Pereda A, O’Brien J, Nagy JI, Bukauskas F, Davidson KG V, Kamasawa N, Yasumura T, Rash JE (2003)

Connexin35 mediates electrical transmission at mixed synapses on Mauthner cells. J Neurosci 23:7489–503.

Rash JE, Curti S, Vanderpool KG, Kamasawa N, Nannapaneni S, Palacios-Prado N, Flores CE, Yasumura T, O’Brien J, Lynn BD, Bukauskas FF, Nagy JI, Pereda AE (2013) Molecular and functional asymmetry at a vertebrate electrical synapse. Neuron 79:957–69.

Rash JE, Kamasawa N, Vanderpool KG, Yasumura T, O’Brien J, Nannapaneni S, Pereda AE, Nagy JI (2015) Heterotypic gap junctions at glutamatergic mixed synapses are abundant in goldfish brain. Neuroscience 285:166–93.

Robertson JD, Bodenheimer TS, Stage DE (1963) The ultrastructure of Mauthner cell synapses and nodes in goldfish brains. J Cell Biol 19:159–99.

Tuttle R, Masuko S, Nakajima Y (1986) Freeze-fracture study of the large myelinated club ending synapse on the goldfish Mauthner cell: special reference to the quantitative analysis of gap junctions. J Comp Neurol 246:202–11.

Wolman MA, Jain RA, Marsden KC, Bell H, Skinner J, Hayer KE, Hogenesch JB, Granato M (2015) A Genome-wide Screen Identifies PAPP-AA-Mediated IGFR Signaling as a Novel Regulator of Habituation Learning. Neuron 85:1200–1211.

Yao C, Vanderpool KG, Delfiner M, Eddy V, Lucaci AG, Soto-Riveros C, Yasumura T, Rash JE, Pereda AE (2014) Electrical synaptic transmission in developing zebrafish: properties and molecular composition of gap junctions at a central auditory synapse. J Neurophysiol 112:2102–2113.

Zhao X, Garcia J, Royer LA, Guo S (2022) Colocalization Analysis for Cryosectioned and Immunostained Tissue Samples with or without Label Retention Expansion Microscopy (LR-ExM) by JACoP. Bioprotocol 12.

Zhao X, Garcia JQ, Tong K, Chen X, Yang B, Li Q, Dai Z, Shi X, Seiple IB, Huang B, Guo S (2021) Polarized endosome dynamics engage cytoplasmic Par-3 that recruits dynein during asymmetric cell division. Sci Adv 7.

Zhu C et al. (2021) Measurement of expansion factor and distortion for expansion microscopy using isolated renal glomeruli as landmarks. J Biophotonics 14:e202100001.